



# Retrieval of Gridded Aerosol Direct Radiative Forcing Based on Multiplatform Datasets

Yanyu Wang[1], Rui Lyu[1], Xin Xie[1], Meijin Huang[2], Junshi Wu[3], Haizhen Mu[3], Qiu-Run Yu[4], Qianshan He[3,5*], Tiantao Cheng[6,7,1*]

[1]Shanghai Key Laboratory of Atmospheric Particle Pollution and Prevention (LAP$^3$), Department of Environmental Science and Engineering, Institute of Atmospheric Sciences, Fudan University, Shanghai, 200438, China
[2]Fujian Meteorological Observatory, Fuzhou, 350001, China
[3]Shanghai Meteorological Service, Shanghai, 200030, China
[4]Key Laboratory of Meteorological Disaster, Ministry of Education (KLME)/Joint International Research Laboratory of Climate and Environment Change (ILCEC), Nanjing University of Information Science and Technology, Nanjing, 210044, China
[5]Shanghai Key Laboratory of Meteorology and Health, Shanghai, 200030, China.
[6]Department of Atmospheric and Oceanic Sciences, Institute of Atmospheric Sciences, Fudan University, Shanghai, 200438, China
[7]Shanghai Institute of Eco-Chongming (SIEC), Shanghai, 200062, China

*Correspondence to*: Qianshan He (oxeye75@163.com); Tiantao Cheng(ttcheng@fudan.edu.cn).

**Abstract.** Atmospheric aerosols play a crucial role in regional radiative budgets. Previous studies on clear-sky aerosol direct radiative forcing (ADRF) have mainly been limited to site-scale observations or model simulations for short-term cases, and long-term distributions of ADRF in China has not been portrayed yet. In this study, an accurate fine-resolution ADRF estimate at the surface was proposed. Multiplatform datasets, including satellite (Terra and Aqua) and reanalysis datasets, served as inputs to the Santa Barbara Discrete Atmospheric Radiative Transfer (SBDART) model for ADRF simulation with consideration of the aerosol vertical profile over East China during 2000-2016. Specifically, single scattering albedo (SSA) from the Modern-Era Retrospective Analysis for Research and Application, version 2 (MERRA-2) was validated with sunphotometers over East China. The gridded asymmetry parameter (ASY) was then simulated by matching the calculated top-of-atmosphere (TOA) radiative fluxes from the radiative transfer model with satellite observations (Clouds and the Earth's Radiant Energy System (CERES)). The high correlation and small discrepancy (6-8 W m$^{-2}$) between simulated and observed radiative fluxes at three sites (Baoshan, Fuzhou, and Yong'an) indicated that ADRF retrieval is feasible and has high accuracy over East China. Then this method was applied in each grid of East China and the overall picture of ADRF distributions over East China during 2000-2016 was displayed. ADRF ranges from -220 to -20 W m$^{-2}$, and annual mean ADRF is -100.21 W m$^{-2}$, implying that aerosols have strong cooling effect at the surface during past 16 years. Finally, uncertainty analysis was also evaluated. Our method provides the long-term ADRF distribution over East China for the first time, with highlighting the importance of aerosol radiative impact under the climate change.



# 1 Introduction

Atmospheric aerosols play a significant role in air quality, regional/global climate and human health (Wang et al., 2018;
Wang et al., 2019). Aerosols can directly absorb and scatter solar radiation, and indirectly affect cloud formation and
precipitation by acting as cloud condensation nuclei or ice nuclei (Twomey, 1977; Rosenfeld, 1999). Large amounts of
scattering aerosols can generally attenuate incoming solar radiation. This reduction in surface radiation significantly impacts
the surface temperature, crop growth and solar energy availability (Chameides, 1999; Liao et al., 2016). On the other hand,
highly absorbing aerosols, such as black carbon, can warm the atmosphere, alter regional atmospheric stability, and even
influence the large-scale circulation and hydrologic cycle with significant regional climate effects (Menon et al., 2002; Wang,
J. et al., 2009). Aerosol direct radiative forcing (ADRF) is a good metric for evaluating the impact of aerosols to radiation by
absorption and scattering, and is defined as the difference between the net radiative flux of earth-atmosphere systems with
and without aerosols. Anthropogenic aerosols produce -0.35±0.5 W m$^{-2}$ of ADRF, which has dampened the warming effect
of greenhouse gases (IPCC,2013). However, the current assessment of ADRF remains highly uncertain. This uncertainty
mainly results from the large variations in aerosol concentrations, chemical compositions, optical properties, mixing states,
and vertical profiles (Haywood and Boucher, 2000; Tian et al., 2018a). Therefore, an accurate and feasible method for
ADRF retrieval is greatly required.

Reduction in these uncertainties requires the integration of different techniques and datasets (e.g., surface measurement,
model simulation, and satellite remote sensing) (Yu et al., 2006). To better understand aerosol optical properties and their
radiative effect, several ground-based networks have been established worldwide, such as the AEROsol Robotic Network
(AERONET) (Holben et al., 2001), Global Atmosphere Watch-Precision Filter Radiometer network (GAW-PFR) (Nyeki et
al., 2005), China Aerosol Remote Sensing Network (CARSNET) (Che et al., 2009) and Chinese Sun Hazemeter Network
(CSHNET) (Xin et al., 2007). Moreover, intensive field experiments have been carried out over China, and these
measurements imply that aerosols exert different levels of cooling effect near the surface in different regions, such as Beijing,
Xianghe, Taihu, Wuhan, Shanghai, Lanzhou (Li et al., 2003; He et al., 2012a; Wang et al., 2014; Yu et al.,2016a; Gong et al.,
2017; Zhang et al.,2018). Such measurements are conducive to wider knowledge of aerosol properties, which are helpful for
improving the performance of satellite and model simulations through synthesis. Nevertheless, the available measurements
usually restricted in terms of spatial and temporal coverage. In addition to surface measurements, model simulations play an
indispensable role in the estimation of the aerosol radiative effect at the global scale and excel in predicting past or future
trends of ADRF (Chang and Liao, 2009; Qiu et al., 2016). Meanwhile, model simulations are subject to large uncertainties in
terms of emissions, transport, and physical and chemical parametrization schemes (José A. et al., 2013).

Compared to the above methods, satellite remote sensing has an outstanding advantage of delivering aerosol information
with high spatial resolution and continuous temporal coverage. Using solely satellite data or a combination with model
simulations and observations constraint, many methods have been developed to retrieve global and regional ADRF estimates
(e.g., Yu et al., 2004; Bellouin et al., 2005; Graaf et al., 2013). However, these studies have mainly concentrated on the top-



of-atmosphere (TOA) radiation budget. Thus far, long-term estimates of the surface ADRF distribution have rarely been addressed, especially in China, one of the most populated and polluted regions globally. This lack of research is because satellites are unable to measure surface-level radiative fluxes directly. Furthermore, aerosol microphysical parameters are crucial in ADRF simulation, including single scattering albedo (SSA, see Table 1 for the acronyms) and the asymmetry

parameter (ASY), but their retrieval remains challenging. Many attempts have been made to solve this key problem. For instance, Thomas et al. (2013) adopted prescribed aerosol properties from the literature to estimate surface ADRF. Fu et al. (2017) took aerosol optical parameters from some AERONET stations as representative of the entire region to conduct grid-cell ADRF simulations. Undoubtedly, additional uncertainty is introduced by the assumption of aerosol optical representativeness in the temporal and spatial dimensions. Some studies also nudged global model simulations towards

AERONET SSA to obtain the aerosol parameters (Chung et al., 2016). With the rapid development of satellite technology, more satellites are providing more detailed aerosol optical products via instruments such as the Polarization and Directionality of the Earth's Reflectance instrument (POLDER), and the Ozone Monitoring Instrument (OMI) (Levet, et al., 2006; Tilstra, et al., 2007). However, the accuracy of the SSA and ASY products over China is still undesirable (Oikawa et al., 2013; Dubovik, et al., 2019). Recently, using satellite and observational data assimilated into the Goddard Earth

Observing System, version 5 (GEOS-5), the National Aeronautics and Space Administration (NASA) has extended the Modern-Era Retrospective Analysis for Research and Application, version 2 (MERRA-2). Compared with its predecessor (MERRA-1), MERRA-2 offers important improvements in aerosol assimilations (Gelaro et al., 2017). The new dataset has the potential to provide improved estimates of aerosol microphysical parameters, such as SSA, and can be further used in the ADRF estimation. After SSA determined, ASY, the only unknow model inputs, can be retrieved by matching the simulated

radiative fluxes with satellite measurements from Clouds and the Earth's Radiant Energy System (CERES). Overall, based on the satellite and reanalysis datasets, including MERRA-2, the MODerate Resolution Imaging Spectroradiometer (MODIS) and CERES, the objective of this study is to propose quantitative estimates of fine-resolution ADRF in the clear sky using a radiative transfer model. Here, East China (114°-124°E, 24°-38°N, shown in the Figure 1) was taken as the validation area of ADRF retrieval, and the simulated radiative fluxes were compared with surface radiation measurements in East China.

Additionally, the aerosol vertical profiles in each grid, which were not considered in previous studies, are used to obtain more accurate ADRF. The data acquisition is presented in Section 2, and Section 3 introduces the method of ADRF simulations. Section 4 including the retrieval of aerosol optical properties, validation of surface radiative fluxes with pyranometers, and detailed discussion of the error sources. Then this method was applied in each grid of East China during 2000-2016, and the uncertainty in the retrieval method also discussed in Section 4. The conclusion is presented in Section 5.

**2 Data**

To acquire ADRF, the inputs (aerosol optical depth (AOD), SSA, ASY, albedo, etc.) to the radiative transfer model were determined from a combination of satellite and reanalysis datasets. AOD was derived from Collection 6 (C6) of MODIS Level 2 products over land (10-km resolution at the nadir) from the Terra satellite (Levy et al., 2013). MODIS/AOD retrieval





primarily employs three spectral channels, centered at 0.47, 0.66, and 2.1 μm and is interpolated at 0.55 μm (Kaufman et al.,
1997). Li et al. (2003) demonstrated that the MODIS/AOD Level 2 product is appropriate in eastern China and exhibits high
precision. In addition, He et al. (2010) found that MODIS/AOD was highly correlated with sunphotometer (CE318)
measurements at 7 sites in the Yangtze River Delta (YRD) region (118°-123°E, 29°-33°N), with a correlation coefficient of
0.85 and with 90% of cases falling in the range of ΔAOD = ± 0.05 ± 0.20 AOD (Chu et al., 2002). Thus, the uncertainty in
the AOD is regarded as 20% in this study.

The hourly SSA product provided by MERRA-2 was estimated by the ratio of total aerosol scattering aerosol optical
thickness (AOT) to total aerosol extinction AOT at a wavelength of 0.55 μm. MERRA-2 combines GEOS-5 and the three-
dimensional variational data assimilation (3DVar) Gridpoint Statistical Interpolation analysis system (GSI). GEOS-5 is
coupled to the Goddard Chemistry, Aerosol, Radiation and Transport (GOCART) aerosol module, which includes five
particulate species (sulfate, dust, sea salt, organic and black carbon) (Colarco et al., 2010). The optical properties of these
aerosols are primarily from the Optical Properties of Aerosols and Clouds (OPAC) dataset (Hess et al., 1998). More details
of the aerosol module in MERRA-2 can be found in Randles et al. (2017) and Buchard et al. (2017). The new dataset has
been used in many recent studies and is appropriate for environmental and atmospheric research (Song et al., 2018).

The upward radiative flux at TOA was used to constrain and determine the ASY. The shortwave (SW, 0.3-5 μm) TOA flux
was acquired by CERES Single Scanner Footprint (SSF) level 2 product from Terra satellite. CERES SSF measures the
instantaneous reflected SW radiance under clear-sky conditions. To convert from radiance to flux, angular distribution
models (ADM) were used in the CERES SSF product (Loeb et al., 2003). The CERES file contains one hour of data, and the
CERES SSF footprint nadir resolution is approximately 20 km. According to Su et al. (2015), the uncertainty of TOA SW
flux is 1.6% over clear land.

Another important parameter for ADRF simulations is the surface albedo. The black-sky albedo, derived from the MODIS
MCD43C3 SW albedo product (C6), was used in this study. Each file contains 16 days of combined Level 3 data from the
satellites Aqua and Terra, with a spatial resolution of 0.5°. Notably, to ensure accuracy, only the albedo values with a high
confidence index were used. The uncertainty in the high-quality MODIS albedo is less than 5% (Cescatti et al., 2012).

The total column ozone, total column water vapor and atmospheric profile data were from the ERA-Interim (European
Center for Medium-Range Weather Forecast (ECWMF) Interim Reanalysis). Specifically, the atmospheric profile includes
the altitude, temperature, water vapor density, and ozone density at 37 pressure levels (1, 2, 3, 5, 7, 10, 20, 30, 50, 70, 100 to
250 at 25-hPa intervals, 300 to 750 at 50-hPa intervals, and 775 to 1000 at 25-hPa intervals). The data quality of the ERA-
Interim reanalysis data can be found in Dee et al. (2011).

The aerosol vertical profile plays a non-negligible role in aerosol radiative forcing. Here, the aerosol vertical profile model
retrieved by He et al. (2016) was applied in each grid to take the place of the default in the radiative model. The retrieval can
130  be briefly described as follows. Based on the two-layer aerosol model, two crucial parameters of the aerosol vertical profile
are the planet boundary layer height (PBL) and the aerosol layer height (ALH) (He et al.,2008). The aerosol extinction
coefficient is assumed to decrease exponentially with altitude above the top of the PBL, and the ALH is defined as the level


where the aerosol extinction coefficient decreases to 1/e (scaling height) of that at the top of the PBL. The PBL was simulated using a three-domain, two-way nested simulation of the WRF Model (Weather Research and Forecasting Model, version 3.2.1). ALH was estimated by the meteorological parameters (relative humidity, temperature, wind speed and wind direction) from the National Centers for Environmental Prediction-Final Operational Global Analysis (NCEP-FNL) via an automated workflow algorithm. The aerosol profiles were utilized to calculate the surface-level visibility from AOD, and the long-term spatial comparison with surface measurements over East China displayed that 90% of the samples exhibited correlation coefficients greater than 0.6 and that 68% of the samples exhibited correlation coefficients greater than 0.7 (He et al., 2016).

All of these multiplatform datasets with their spatial and temporal resolutions are summarized in Table 2. In this study, these datasets were interpolated to a spatial resolution of 0.1°×0.1° to collocate with the MODIS/AOD data. Additionally, the ADRF simulation was performed in each 0.1°×0.1° grid over East China. The temporal coverage is from 2000 to 2016. The research area and surface measurement sites for validation are shown in Figure 1.

## 3 Methodology

Clear-sky ADRF in the SW (0.25-4 μm) spectral region was simulated by the Santa Barbara Discrete Atmospheric Radiative Transfer (SBDART) model (Ricchiazzi et al., 1998). This model has been widely adopted for the estimation of aerosol radiative forcing and validated with high accuracy (Li et al., 2010). As shown in Figure 2, the main inputs of the SBDART model include aerosol properties (AOD from MODIS; SSA from MERRA-2; ASY from the retrieval (Section 4.2)), surface albedo (from MODIS), aerosol vertical profile (from NCEP), atmospheric profiles (from ECWMF), total column ozone and water vapor (from ECWMF). The main outputs are radiative fluxes at the surface and TOA with and without aerosols. ADRF is defined as the difference in net radiative flux (downward minus upward) between aerosol and no-aerosol conditions. Here, we mainly concentrated on ADRF at the surface:

$$\text{ADRF}_{\text{sur}} = (F\downarrow - F\uparrow) - (F_0\downarrow - F_0\uparrow), \tag{1}$$

where $F$ and $F_0$ represent radiative fluxes with and without the aerosol at the surface, respectively. The upward and downward arrows denote the directions of the radiative fluxes, which can be obtained by the outputs of SBDART. For simplicity, the upward radiative fluxes at the TOA are called F_u_toa, and the downward/upward radiative fluxes at the surface are called F_d_sur and F_u_sur, respectively (see Table 1 for the acronyms).

## 4 Results and discussion

### 4.1 Retrieval of aerosol properties

Before ADRF simulation, one of the inputs, SSA from MERRA-2, was evaluated firstly. In East China, three sunphotometer sites, Pudong (121.79ºE, 31.05ºN), Taihu (120.22ºE, 31.42ºN), and Xuzhou (117.14ºE, 34.22ºN), were chosen for comparison with MERRA-2 SSA data due to their large available samples, while other sites in East China did not have



enough data for analysis. The blue triangles in Figure 1 represent the locations of the sunphotometers, and their geographical

characteristics and observation periods can be found in Table 3. Taihu and Xuzhou are AERONET sites and Level 1.5 inversion data in AERONET sites were used. The uncertainty in the AERONET products can be found in Dubovik and King (2000). Another sunphotometer (CE318, Cimel Electronique, France) in Pudong is calibrated annually and maintained routinely, and a detailed description of calibration was presented in Cheng et al. (2015). The sunphotometer spectral products are available at wavelengths of 440, 675, 870, and 1020 nm, and they were interpolated at 0.55 μm to match MERRA-2 SSA.

The collection time was constrained from 09:00 to 14:00 (local time), covering the overpass time of the Terra satellite. Meanwhile, the relatively high solar zenith in this period avoids possible inversion errors and improves the data accuracy (Tian et al., 2018b). Additionally, the specific MERRA-2 grid cell containing the sunphotometer was selected, and the sunphotometer SSA was hourly averaged to match the MERRA-2 SSA product. Figure 3 displays the detailed comparisons at Pudong, Taihu, and Xuzhou. The blue solid line represents the fitting curve of the scatter dots, and the dashed lines are the

range of ±10% relative error. All samples in Taihu and Pudong and 94% of samples in Xuzhou fall within the ±10% error. This finding suggests that MERRA-2 SSA agrees well with the sunphotometer data, even though some dots in Xuzhou are beyond the error range. The further comparison between MERRA-2 SSA and sunphotometer are shown in Figure S1 (Supplementary Information). The boxplots for the three sites indicates the mean value of MERRA-2 SSA is similar to previous measurements in East China, such as Shanghai (0.91), Taihu (0.91) and Huainan (0.89), (Liu et al.,2012; Che et

al.,2017; Che et al., 2019). Furthermore, it also reveals that MERRA-2 generally produces lower SSA than surface measurements in Taihu and Xuzhou. The primary reason for the discrepancy between MERRA-2 and surface measurements is the simple aerosol model assumption in MERRA-2 (Buchard et al., 2018). Only five aerosol types (sulfate, dust, sea salt, organic and black carbon) are involved; the lack of nitrate aerosols, which are highly scattering aerosols, is responsible for the underestimation of MERRA-2 SSA, especially in Xuzhou, with various aerosol sources related to human activities (Che

et al., 2015). In addition, the calibration errors among the three instruments should be considered. Generally, the evaluation results in three sites show that the accuracy of MERRA-2 SSA product is acceptable in East China, with ±10% uncertainty.

After SSA was determined, ASY is the only unknown model input parameter. ASY is the key to portraying the scattering direction of aerosols. ASY=1 denotes completely forward scattering, and ASY=0 is symmetric (Rayleigh) scattering. Here, gridded ASY was simulated by matching observed F_u_toa from CERES with simulated F_u_toa from SBDART. The

sensitivity test indicates that F_u_toa, just like F_u_sur (shown in Figure 7b), is a monotonically increasing function of ASY with other fixed inputs. Consequently, only one F_u_toa can be obtained by one specific ASY. In this premise, a binary search was applied to approximate ASY to improve calculation efficiency (Chang, 2013). The goal of the binary search is to find the ASY when the simulated F_u_toa is close to the observed F_u_toa. To accomplish this, the ranges of F_u_toa are repeatedly diminished by taking the middle ASY as one of the boundary values, and when the difference between the

F_u_toa observed by CERES and calculated by SBDART is less than 1, the corresponding approximation of ASY is finally obtained. The detailed scheme is illustrated in Figure 4. First, the value for ASY is initially assumed in the reasonable range of 0.1-0.9, and the upper and lower boundaries of ASY, along with other parameters, are input to SBDART to yield the



initial range of calculated F_u_toa_a and F_u_toa_b. Then, this range is checked to determine whether it includes the F_u_toa (observed by CERES) by multiplying ((F_u_toa_a- F_u_toa)*( F_u_toa_b- F_u_toa)). If the multiplication result is

negative, meaning that ASY falls within this range (ASYa, ASYb), the average of F_u_toa_a and F_u_toa_b is set as a new boundary (F_u_toa_c). Otherwise, this case is discarded, and the retrieval is not continued (ASY=NaN), perhaps due to inappropriate inputs. Next, for cases in which the multiplication result is negative, the multiplication process is applied to the new boundary ((F_u_toa_a- F_u_toa)*( F_u_toa_c- F_u_toa)). If this multiplication result is negative, the ASY falls within this range (ASYa, ASYc). Then, ASYc is set to represent ASYb. Otherwise, ASYc is set to represent ASYa. This process

represents the scope-narrowing of the ASY boundary discussed above. With several iterations of narrowing the scope, the boundaries of the simulated F_u_toa become close to the true value of F_u_toa (observed by CERES). When the difference between the simulated F_u_toa boundary and the observed F_u_toa is less than 1, the corresponding ASY is considered as one approximation. In this process, the input parameters, including AOD (from MODIS), SSA (from MERRA-2), surface albedo (from MODIS), aerosol vertical profile (from NCEP), atmospheric profiles (from ECWMF), total column ozone and

water vapor (from ECWMF), were input into the SBDART model together in every iteration. All these inputs from 2000-2016 were used to simulate ADRF in each grid of East China. All calculations were performed on the Linux system.

After aerosol optical properties were obtained, these parameters from multiplatform datasets can input into the SBDART model to simulate surface radiative fluxes and ADRF in East China according to the methodology in Section 3.

**4.2 Validation of the method**

Before conducting ADRF simulation in each grid of East China during 2000-2016, this method was first to applied in the single grid to assess the performance of ADRF retrieval. Three radiation stations in Baoshan (121.45°E, 31.4°N), Fuzhou (119.29°E, 26.08°N), Yong'an (117.37°E, 25.98°N) was chosen to make the comparisons between calculated F_d_sur and surface observation by the pyranometers (FS-S6, China) during 2014-2016. Red circles in Figure 1 denote the specific locations of pyranometers. Their geographical characteristics and observing periods are listed in Table 3. These

pyranometers had regular maintenance and were calibrated annually through intercomparisons with the basic-reference station. Additionally, quality control has been performed at these sites. The uncertainty in the pyranometers is expected to be 5% (Song, 2013). Simulated F_d_sur was averaged in the scope of a 40 km side length with the center at the pyranometer, and the measured F_d_sur was averaged within ±30 min of the satellite overpass (Ichoku, et al., 2002).

Figure 5 displays the comparison results between simulated F_d_sur and observed F_d_sur by pyranometers at the three sites.

The simulated F_d_sur is fairly consistent with the observations, with correlation coefficients of 0.87 in Baoshan (Figure 5a) and Fuzhou (Figure 5b) and 0.90 in Yong'an (Figure 5c). Root mean squared error (RMSE) is a good indicator for measuring the discrepancy between observed and simulated F_d_sur data. The RMSE is 7.9 W m$^{-2}$ in Baoshan, 7.5 W m$^{-2}$ in Fuzhou and 5.6 W m$^{-2}$ in Yong'an. This discrepancy only accounts for 3-5% of the ADRF, indicating that this retrieval method has a relatively higher accuracy than those in other studies (e.g., Thomas et al., 2013; Fu et al., 2017). Additionally,

all slopes are less than 1, which implies that the method has systematic biases at these sites. That is, the simulated F_d_sur is



overestimated relative to observations in clear conditions but underestimated in polluted conditions. Thus, in very clear or polluted conditions, this method can smooth F_d_sur variations. A similar tendency was found in the comparison between MODIS AOD and sunphotometers in East China by He et al. (2010); therefore, the main systematic error in the ADRF simulation may come from the MODIS AOD. Additionally, all intercepts of the fitting lines are greater than 0, indicating
that the method can produce errors, especially in clear conditions. Nevertheless, satisfactory comparison results indicate the suitability and feasibility of ADRF retrieval in the southern and northern sites of East China, although the types of underlying surface and aerosol sources in the north are evidently different from those in the south.

To further assess the discrepancy between simulated F_d_sur and the observations, the relative errors of each case at the three sites were calculated. The results suggest that underestimated cases (negative relative errors) account for 61% of the
total cases and overestimated cases (positive relative errors) account for 39%. According to the validation results, the sources of error in the simulation may be attributed to the following reasons:

**Cloud contamination:** An examination of cloudiness was carried out at the three sites. According to the empirical clear-sky detection method, one-hour radiation data of a pyranometer was used to discriminate clear-sky observations (Xia et al., 2007). The red dots in Figure 5 represent the cloudiness case detected by the pyranometer. Meanwhile, from the MODIS true
color map composed by channels 1, 4 and 3 (not shown), the olive green dots denote the specific case in which the site is completely covered by clouds. A large discrepancy in these cases between simulated F_d_sur and observation is also evidence of substantial errors produced by clouds. The cloud effect, especially that of residual thin cirrus clouds, is difficult to completely remove from MODIS AOD (Kaufman et al., 2005). Moreover, the cloud mask algorithm in MODIS aerosol inversion sometimes fails to distinguish fog or haze in high-humidity conditions. Many more fog days are observed in
Fuzhou than at the other two sites, and fogginess can significantly reduce the accuracy of the simulation (Ye et al. 2010). In addition, the error source of MODIS AOD is also from errors in the aerosol model assumption and surface reflectivity (Xie et al., 2011).

**Different spatial and temporal representativeness:** In the validation, the area measurement (satellite and reanalysis data) was compared to point measurements (pyranometer). For temporal matching, the pyranometer can capture the process of
perturbation induced by air mass movement within one hour, whereas satellite can only provide the instantaneous conditions. Hence, this comparison method inevitably introduces some degree of uncertainty.

**Instrument and radiative transfer errors:** One error source in pyranometers is the thermal offset effect. This spurious signal is due to the difference in temperature between the inner dome and the detector of a pyranometer and can lead to additional errors in the irradiance measurements, especially diffuse irradiance (Sanchez et al., 2015). To reduce this effect, a
pyranometer should be installed in a transparent ventilation hood. Alternatively, several statistical methods have also been proposed to suppress the thermal offset effect (e.g., Song, 2013; Cheng et al., 2014). In this study, the correction of the thermal offset was not performed because of the lack of additional observation data. Aside from the instrument error, the model simulation discrepancy also depends on the radiative transfer models. They are based on some simplifications, including the sphericity of aerosol particles and the directional reflectance of the surface. Derimian et al. (2016) found that





neglecting aerosol particle nonsphericity can overestimate the aerosol cooling effect. Furthermore, simulation results vary slightly among different models due to their different assumptions in radiative transfer. For instance, Yu et al. (2007) compared three models (second simulation of the satellite signal in the solar spectrum (6S), MODerate resolution atmospheric TRANsmission (MODTRAN) and SBDART) at Xianghe station and showed that approximately 80% of the cases simulated by SBDART were lower than the surface observations, while the 6S simulation results were higher.

**The effect of aerosol sources:** The Hybrid Single Particle Lagrangian Integrated Trajectory (HYSPLIT) model was employed for the backward trajectory of air mass (http://ready.arl.noaa.gov/HYSPLIT.php) to explore the effect of air mass origin on the ADRF simulation. Here, archive data from the Global Data Assimilation System (GDAS) were applied in this model. A 48 h backward trajectory of air mass ending at the three pyranometers at a height of 0.5 km was used to trace the origin of the surface-level air mass. In Fuzhou, almost all the directions of blue lines (Figure S2), which denote negative

relative errors of simulation, are northward, while the directions of red lines with positive errors are southward. The major aerosols associated with the blue lines are inferred to be anthropogenic and high-scattering particles. MERRA-2 SSA is always underestimated in these conditions, potentially leading to the negative errors in the simulated $F\_d\_sur$ because SSA has the same phase as $F\_d\_sur$ (Figure 7a, shown below). Moreover, the direction of the air mass trajectory is found to be steady on consecutive days, and the change in the error sign is consistent with the change in the trajectory direction. Taking

Yong'an as an example, three 48 h backward trajectories of air masses with negative errors all come from northeast during October 22-24, 2015 (Figure S3). This pattern is due to the similar aerosol types accompanying the same weather system over this region. In general, the aerosol source determines the dominant aerosol types and SSA, further producing additional uncertainty in the ADRF simulation.

### 4.3 Long-term ADRF retrieval in East China

The above evaluations show the method for ADRF simulation is feasible and high-accuracy in East China, thus this method was further applied in each grid of East China to obtain a full coverage of ADRF from 2000-2016. Figure 6a outlines an overall picture of annual mean ADRF at the surface over East China during past 17 years. It provides valuable information about aerosol radiative effect not only in the urban areas with intensive human activities, but also in the suburb with unavailable observational data. ADRFs in all grids are negative, ranging from -220 W m$^{-2}$ to -20 W m$^{-2}$, implying that

aerosols have cooling effect on surface over East China. Obvious difference of ADRF distributions is found between the northern and southern part of East China, and the magnitude of ADRF increases from South to North. This pattern is consistent with site observations in Che et al., (2019), in which surface ADRF ranges from -150 to -100 W m$^{-2}$ in the northern sites of East China (Huainan and Hefei) while ADRF ranges from -100 to -50 W m$^{-2}$ in the southern sites of East China (Jiande, ChunAn and Tonglu). Red area denotes the high absolute value of ADRFs (Figure 6a), which are found in the

densely populated and industrialized areas, including the western Shandong, YRD and Poyang Lake Plain. Low value (blue area) is observed in the Southern part, such as Fujian and southern Zhejiang. This pattern is mainly attributed to the difference of industry locations and topography between the North and South. Large amounts of anthropogenic aerosols in





the North are highly scattering, they can impose strong cooling radiative effect. It is worth noting that, although western Shandong has lower urbanization compared with YRD, this cooling effect in western Shandong is even larger than in YRD. This is because Yimeng mountain locates in the middle of Shandong, it blocks the west flow and make the aerosol accumulations near its western border (He et al., 2012b). Meanwhile, Shandong is also easily impacted by air pollution transported from North China. In addition, high absolute value of ADRF is also found in Poyang Lake in Jiangxi with abundance of anthropogenic aerosols, and these areas are surrounded by the mountains, the poor ventilation condition leads to aerosols enhanced. Compared with the North, the South is characterized by more extensive vegetation coverage and less human activities, dominated natural aerosols have weaker cooling effect. The ADRF distribution over East China is similar with AOD, which is presented in He et al. (2012b), that is, the areas of high AOD is corresponding to high value of ADRF. Meanwhile, ADRF also depends on the aerosol types. In some regions of East China with abundant of absorbing aerosols, the positive value of ADRF can occur especially in the bright surface (Sundström et al., 2015). In addition, the temporal variation of ADRF distributions further indicates it changes remarkably in East China over past decades, and the North experiences more notable changes of ADRF compared the South, which needs to be further identified and explored with additional measurements. Figure 6b displays the yearly regional mean changes of ADRF from 2000 to 2016 and the yearly mean ADRF is -100.21 W m$^{-2}$. It reflects ADRF shows a fluctuation pattern, with the lowest, -121.78 W m$^{-2}$ in 2013 and the highest, -93.87 W m$^{-2}$ in 2009. The magnitude of ADRF is higher than the most cities in the world, such as Spain (Esteve et al., 2014), Gasan (Kim et al., 2006) and Karachi (Alam et al., 2011). In addition, aerosol cooling radiative effect can sharply increase with large aerosol loadings. According to Yu et al. (2016b), surface ADRF can reach up to -163 W m$^{-2}$ in the haze days, while in the non-haze days, it can decrease to -45 W m$^{-2}$ in Beijing on January 2013. Usually in the heavy haze, the enhanced surface cooling, with combining of atmosphere heating, can result in a more stable environment, which is unfavourable for the diffusion and dispersion of the aerosols and further exacerbates air pollution (Wu et al., 2019). Therefore, aerosol radiative feedback plays a vital role in the severe haze events in winter.

**4.4 Sensitivity test and uncertainty analysis**

To determine the uncertainty of the method for ADRF simulation caused by each input parameter, a sensitivity test for input parameters was carried out. A specific case in Shanghai on October 11, 2015, was used with the following values: AOD = 0.62, SSA = 0.85, ASY = 0.69, surface albedo = 0.13, total column water vapor = 0.69 g/cm$^2$, and total column ozone = 0.28 atm-cm. Figure 7 portrays the responses of F_d_sur, F_u_sur and ADRF to changes in one parameter while holding the other parameters constant. To remove the impact of units, all the parameters are dimensionless; that is, the ratio of the input to the actual value is used as the x-axis value. The absolute value of every slope describes the impact of every parameter on the dependent variables (F_d_sur, F_u_sur and ADRF). Figure 7 presents the actual condition of this case when the value of the x-axis equals 1, in which F_d_sur is 629.15 W m$^{-2}$, F_u_sur is 83.52 W m$^{-2}$, and ADRF is -149.39 W m$^{-2}$. This situation denotes a strong cooling effect of aerosols at the surface. Apparently, different parameters impose diverse influences on the radiative values (F_d_sur, F_u_sur, and ADRF). As depicted in Figure 7a, AOD, SSA, and ASY are three crucial parameters





that greatly influence F_d_sur. For F_u_sur, albedo, AOD, and SSA are more important parameters (Figure 7b). The impact of surface albedo is much larger than the others because albedo actually determines how much of the irradiance is reflected by the surface. Figure 7c implies that SSA, AOD, and ASY are major factors in determining ADRF. Additionally, only a large AOD produces much cooler at the surface, whereas increases in SSA and ASY can result in decreases in the aerosol
cooling effect. In general, sensitivity test shows that ADRF depends highly on AOD, SSA, ASY and albedo. Two parameters (atmospheric profile and aerosol vertical profile) are not discussed because these parameters have little impact on clear-sky ADRF in the above case. The atmospheric profile has a minor effect on the perturbations of ADRF compared with the total columns of atmospheric component (water vapor and ozone). This result has also been proven by Yu et al. (2007) and Li et al. (2016). The sensitivity test also shows that, with a fixed total column of AOD, clear-sky ADRF is not sensitive to the
shapes of aerosol profiles. However, this effect becomes much stronger in the presence of absorbing aerosols, especially in some extreme cases such as dust storms and biomass burning (Wang and Christopher, 2006; Guan et al., 2009). Reddy et al. (2013) also demonstrated that surface aerosol radiative forcing can be enhanced by 25% due to the insertion of the extinction profile of absorbing aerosols to replace the default profile.

On the basis of these four high-sensitivity factors, the uncertainties in ASY and ADRF due to these parameters were
quantitatively assessed. According to data uncertainty mentioned in Section 2 and the validation result of SSA, the relative errors of AOD, SSA, albedo, and CERES F_u_toa are 20%, 10%, 5% and 1.6%, respectively. This lower/upper limit of parameter errors was input to the ADRF calculation, and the associated uncertainty was calculated by the difference between the simulated radiative flux with parameter errors and without errors. Notably, the uncertainty analysis is based on extreme conditions, and the associated values are much larger than the actual values. As displayed in Table 4, the uncertainty in ASY
induced by SSA can reach up to 23%, indicating that SSA is a decisive factor in ASY retrieval when using the CERES F_u_toa constraint. SSA also has the largest effect in regulating aerosol radiative forcing, which is consistent with the research on dust aerosols by Huang et al. (2009). AOD contributes uncertainties of 3.7% in ASY and 15.4% in ADRF. Albedo introduces 1.7~3.7% uncertainty in ASY and approximately 3% in ADRF. The error of the CERES product produces approximately 1.7% uncertainty in ASY and 1.5% in ADRF. The results of the uncertainty analysis are similar to those of
previous studies. For example, Xia et al. (2016) revealed that AOD and SSA together can account for 94% of the surface ADRF. Zhuang et al. (2018) further noted that the error sources from the absorbing component of AOD and coarse-aerosol SSA contributed to the greater uncertainty in the ADRF. Therefore, improving the precision of the input parameter is helpful for obtaining reliable ADRF estimation, especially in the surface (Wang, P., et al.,2009). As Michalsky et al. (2006) demonstrated, when using high-quality measurements as inputs to model, the biases between modeled and measured
irradiance can decrease to 1.9%. In addition to these factors, Wang and Martin (2007) also revealed the effects of aerosol hygroscopicity on the aerosol phase function and the increase in SSA with RH enhancement, suggesting that relative humidity (RH) is also closely related to ADRF.





## 5 Conclusion

In this study, based on multiplatform datasets, high-accuracy ADRF distributions over East China during 2000-2016 were
determined. MERRA-2 SSA data were first compared with sunphotometer data (Taihu, Xuzhou, Pudong), and the validation
result shows that the relative error of the MERRA-2 SSA is ±10% over East China. Then, ASY in each grid was retrieved by
matching the simulated F_u_toa by SBDART with observations based on the CERES product. A binary search was used in
ASY retrieval to improve the retrieval efficiency. Then, aerosol optical properties (AOD from MODIS, SSA from MERRA-
2, and ASY from the retrieval), surface albedo (from MODIS), aerosol vertical profile (from NCEP), atmospheric profiles
(from ECWMF), total column ozone and water vapor (from ECWMF) served as input parameters for SBDART to simulate
ADRF in each grid cell of East China during 2000-2016.

The validation result of this method at three sites (Baoshan, Fuzhou, and Yong'an) reveals that simulated F_d_sur is highly
correlated with the pyranometer data during 2014-2016, with correlation coefficients of 0.87 in Baoshan and Fuzhou and
0.90 in Yong'an. The RMSEs are 7.9 W m$^{-2}$ in Baoshan, 7.5 W m$^{-2}$ in Fuzhou and 5.6 W m$^{-2}$ in Yong'an, showing that
ADRF retrieval is feasibile and has high accuracy over East China. Furthermore, the simulation is found to have systemic
errors at all sites and that it is overestimated in clear conditions and underestimated in polluted conditions. This pattern is
similar to the validation of MODIS AOD with sunphotometers over East China and indicates that the major error source in
ADRF simulations possibly comes from MODIS AOD inversion. In addition, associated factors, including cloud
contamination, instrument and radiative transfer errors, as well as different spatial and temporal representativeness, were
confirmed to produce additionally uncertainty in ADRF simulations. Further analysis of the air mass origin also
demonstrates that ADRF is closely related to the aerosol types and SSA.

After validation this method in three sites, ADRF simulation was conducted in each grid of East China during 2000-2016.
Long-term ADRF distribution over East China was portrayed for the first time. ADRFs in all grids are negative, the range of
ADRF is between -220 W m$^{-2}$ and -20 W m$^{-2}$, implying that aerosols have cooling effect on surface over East China. The
yearly regional mean ADRF is -100.21 W m$^{-2}$. It reflects ADRF shows a fluctuation pattern, with the lowest, -121.78 W m$^{-2}$
in 2013 and the highest, -93.87 W m$^{-2}$ in 2009. The magnitude of ADRF is higher than the most cities in the world, such as
Spain (Esteve et al., 2014), Gasan (Kim et al., 2006) and Karachi (Alam et al., 2011). Obvious difference of ADRF
distributions is found between the northern and southern part of East China. ADRF distribution is similar to AOD pattern in
East China presented in He et al. (2012b). This pattern is mainly attributed to the difference of industry locations and
topography between the North and South. Finally, sensitivity test shows that ADRF depends highly on AOD, SSA, ASY and
albedo. Uncertainty analysis shows the uncertainty in ADRF retrieval induced by SSA is calculated 24% and that by AOD is
15.4%.

In summary, this study suggests that the method for ADRF retrieval can be utilized over the areas with large variations in
aerosol loadings and surface properties. Especially in suburbs with no monitoring resources, our study offers valuable
information on the direct radiative impact of aerosols. It is noted that, in our study, ADRF was calculated during the time





that satellite passes by rather than the whole day. Furthermore, aerosol optical parameters, including AOD and SSA, were considered only at 0.55 μm, and multi-wavelength of them can input to the radiative transfer model to improve the ADRF accuracy (Wang, P., et al., 2009). More additional observation data from the sites, are needed to further verify the performance of the ADRF retrieval and constrain these multiplatform datasets to improve the ADRF accuracy. In addition, it is necessary to improve the satellite instruments and the retrieval algorithm of aerosol properties; more novel methods, such as machine learning, can be involved in the ADRF estimates (Yin, 2010; Yu and Song, 2013). In the future work, the aerosol-induced changes in the surface radiation under climate change and agricultural economic impact also will be studied. This work will provide a deep understanding of aerosol radiative effects and is also helpful for aerosol modeling over East China.

*Data availability.* AOD from MODIS is available at http://ladsweb.nascom.nasa.gov/data/search.html, albedo is also from MODIS (https://e4ftl01.cr.usgs.gov/MOTA/MCD43C3.006/). SSA from MERRA-2 is available at https://disc.gsfc.nasa.gov/daac-bin/FTPSubset2.pl. TOA flux is from CERES (https://ceres.larc.nasa.gov/products.php?product=SSF-Level2). Atmospheric aerosol profile is retrieval from NCEP/NCAR (http://www.esrl.noaa.gov/psd/data/gridded/data.ncep.reanalysis.html). Total column ozone, total column water vapor and atmospheric profile are from ECWMF (https://www.ecmwf.int/en/forecasts/datasets/reanalysis-datasets/era-interim). The SSA from AERONET sites are available at http://aeronet.gsfc.nasa.gov/. HYSPLIT trajectory and dispersion model is simulated at http://ready.arl.noaa.gov/HYSPLIT.php.

*Competing interests.* The authors declare that they have no conflict of interest.

*Author contribution.* Qianshan He and Yanyu Wang designed and conducted the research and analysis. Rui Lyu, Xie Xin and Tiantao Cheng contributed to data analysis and interpretation. Meijin Huang and Junshi Wu provided the surface measurements data. Haizhen Mu offered the computational resources. Qiu-Run Yu collected the reanalysis datasets. Yanyu Wang wrote the manuscript. All authors contributed to improve the manuscript.

*Acknowledgement.* This study was supported by the National Natural Science Foundation of China (41775129 and 91637101), the China National Key Research and Development Plan (2016YFC0202003, 2017YFC1501405, and 2017YFC1501701), and the Science and Technology Commission of Shanghai Municipality (16ZR1431700). We express our great appreciation to all the staffs in Shanghai and Fujian Meteorological Service for establishing and maintaining the observation sites. The Principal Investigators of the AERONET sites are appreciated for providing data on aerosol properties.



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



**Table 1: Summary of the acronyms.**

| | |
|---|---|
| ADRF | Aerosol direct radiative forcing (W m$^{-2}$) |
| SSA | Single scattering albedo (unit less) |
| ASY | Asymmetry parameter (unit less) |
| AOD | Aerosol optical depth (unit less) |
| F_u_toa | Upward radiative fluxes at the top of atmosphere (W m$^{-2}$) |
| F_d_sur | Downward radiative fluxes at the surface (W m$^{-2}$) |
| F_u_sur | Upward radiative fluxes at the surface (W m$^{-2}$) |




**Table 2: Satellite and reanalysis datasets used in the study.**

| Parameters | Products | Sensors/Models | Spatial Resolution | Temporal Resolution |
|---|---|---|---|---|
| AOD | MOD04 L2 | Terra MODIS | 0.1°×0.1° | instantaneous |
| SSA | tavg1_2d_aer_Nx | MERRA-2 | 0.625°×0.5° | hourly |
| Surface albedo | MCD43C3 | Terra+Aqua MODIS | 0.2°×0.2° | daily |
| Upward TOA radiative flux | SSF | Terra CERES | 20km | instantaneous |
| Meteorological data | ERA-Interim | ECMWF | 0.125°×0.125° | hourly |


**Table 3: The geographical characteristics of observation sites for sunphotometer and pyranometer.**

| Location | Lon/Lat | Instrument (Product) | Observing Period |
|---|---|---|---|
| Pudong (Urban) | 121.79°E/31.05°N | Sunphotometer (SSA) | 2010.12-2012.10 2014.1-2015.11 |
| Taihu (Rural) | 120.22°E/31.42°N | Sunphotometer (SSA) | 2005.1-2012.12 2015.1-2016.12 |
| Xuzhou (Urban) | 117.14°E/34.22°N | Sunphotometer (SSA) | 2013.8-2016.12 |
| Baoshan (Urban) | 121.45°E/31.4°N | Pyranometer (F_d_sur) | 2014.1-2016.12 |
| Fuzhou (Urban) | 119.29°E/26.08°N | Pyranometer (F_d_sur) | 2014.1-2016.12 |
| Yong'an (Rural) | 117.37°E/25.98°N | Pyranometer (F_d_sur) | 2014.1-2016.12 |



**Table 4: Errors induced by different input parameters in ASY, radiative flux (F_d_sur, F_u_sur) and ADRF. Here, the uncertainties of input parameters (AOD, Albedo, CERES F_u_toa) are from literatures and the uncertainty of SSA is from validation in Section 4.**

| Parameter | Uncertainty | Errors in ASY | Errors in F_d_sur | Errors in F_u_sur | Errors in ADRF |
|---|---|---|---|---|---|
| AOD | $\pm 20\%$[a] | -3.7%~1.7% | ~4.5% | ~4.4% | ~15.4% |
| SSA | $\pm 10\%$ | -19%~23% | ~12% | ~12% | ~24% |
| Albedo | $\pm 5\%$[b] | -3.7%~1.7% | ~0.7% | ~5.9% | ~3% |
| CERES F_u_toa | $\pm 1.6\%$[c] | -1.8%~1.7% | ~0.4% | ~0.4% | ~1.5% |

[a] He et al. (2010).

[b] Cescatti et al. (2012).

[c] Su et al. (2015).

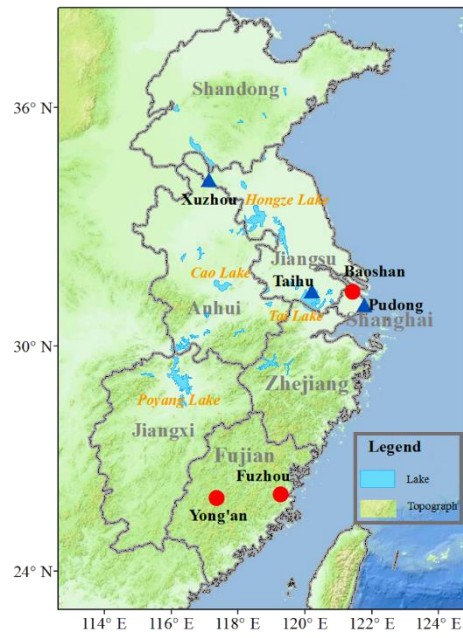

**Figure 1: The map of research area, major lakes and topography in East China are shown. The blue triangles denote the locations of sunphotometers and the red circles are pyranometers. This figure was generated by ArcGIS, version 10.2. Map source: Map World (National Platform for Common Geospatial Information Services, www.tianditu.gov.cn/) .**




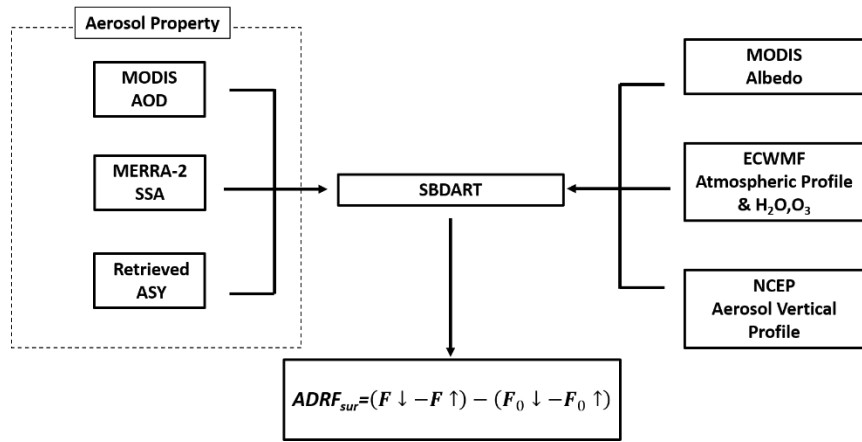

**Figure 2: A schematic diagram to simulate ADRF based on satellite and reanalysis datasets.**


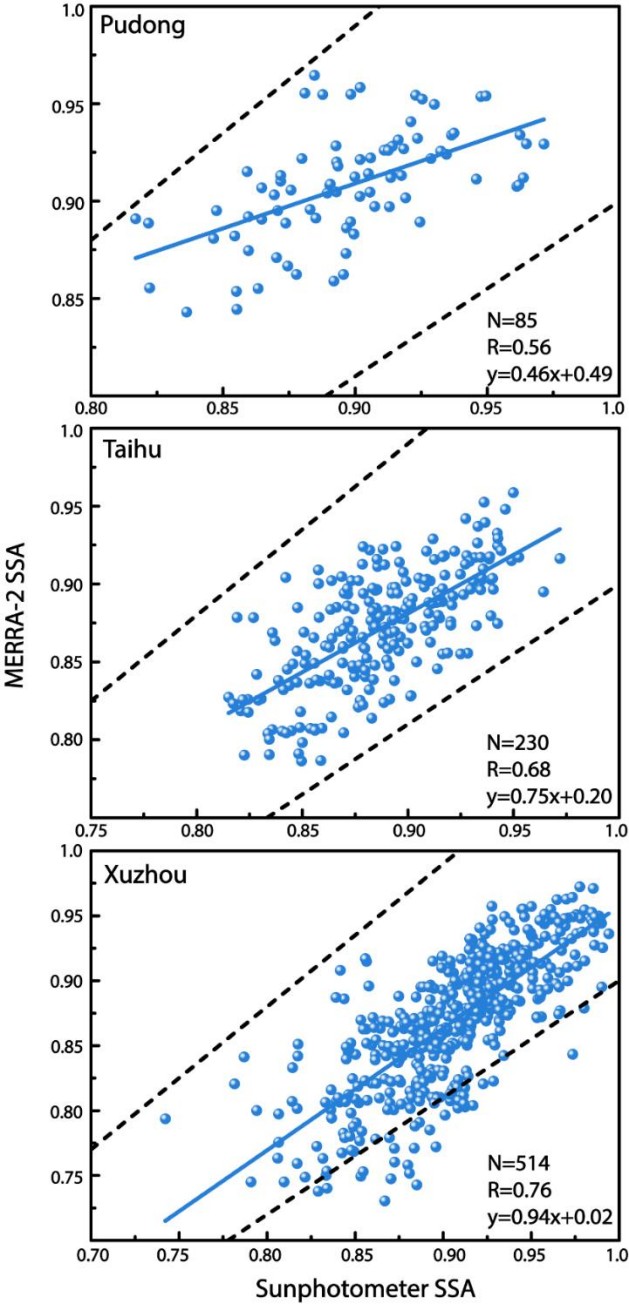

**Figure 3: The scatter plots of SSA between MERRA-2 and sunphotometer in Pudong, Taihu, and Xuzhou. The blue line is the fitting curve while dashed lines are the range of ±10% relative error.**






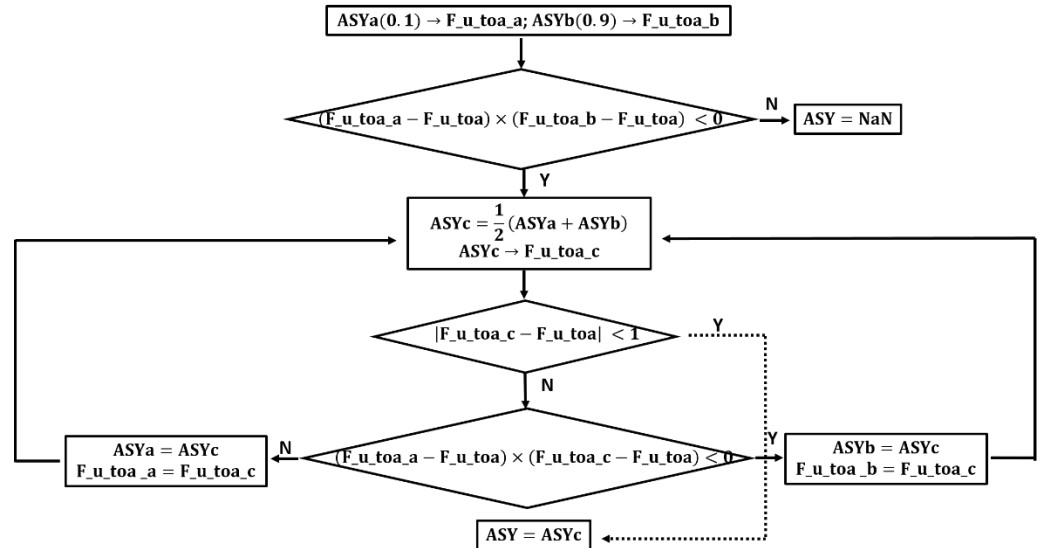

**Figure 4: A detailed workflow of binary search used in ASY retrieval.**

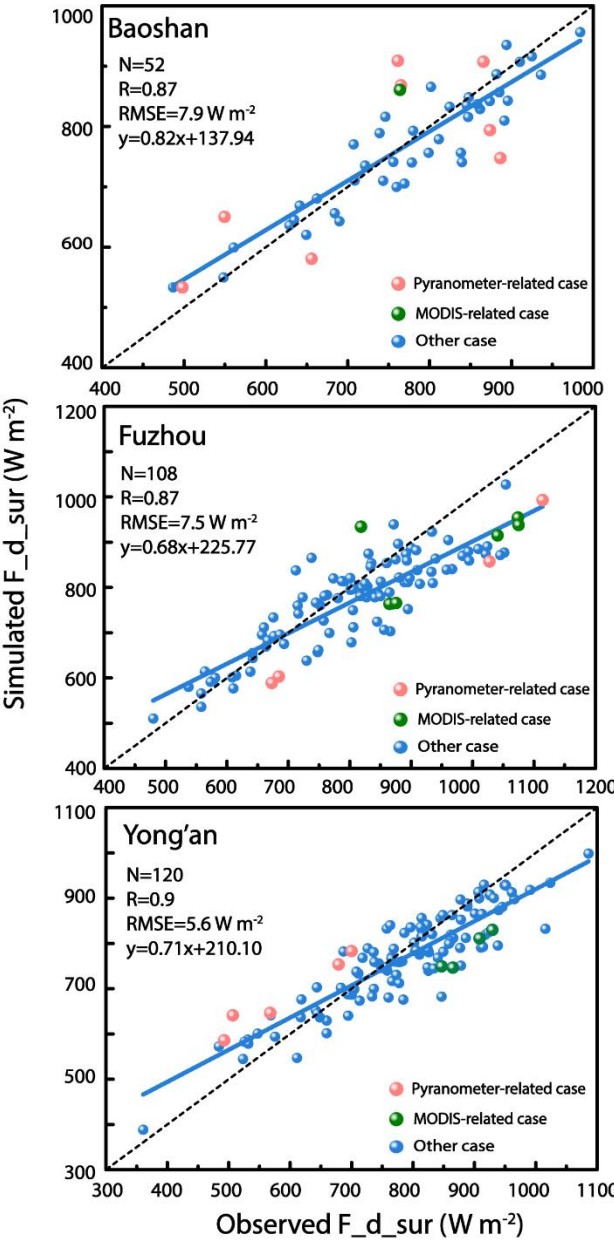

**Figure 5: The scatter plots between observed F_d_sur by pyranometers and simulated F_d_sur by SBDART in Baoshan, Fuzhou, and Yong'an. The blue line the is fitting curve and the dashed line represents y=x. The red dots denote the specific case in which the pyranometer captures the fluctuation of F_d_sur by clouds during one hour. The olive green dots denote the specific case in which the site is completely covered by clouds, deduced from MODIS true color map composed by 1, 4 and 3 channels. The blue dots represent the other ordinary case.**






**Figure 6: (a)Annual mean ADRF distributions during 2000-2016 over East China. (b)The changes of annual regional mean ADRF during 2000-2016 over East China.**





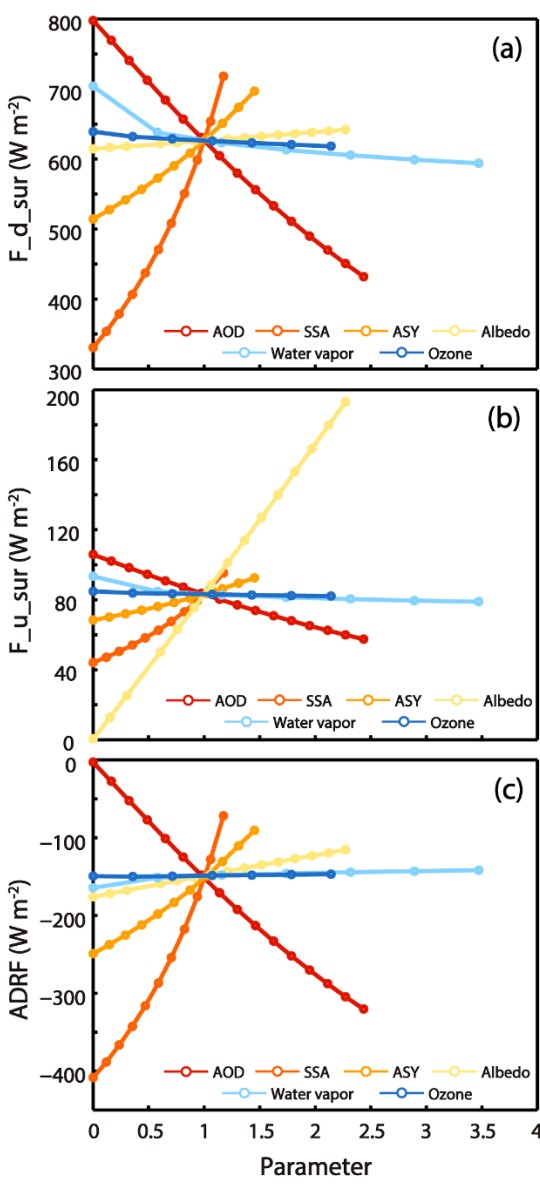

**Figure 7: The response of F_d_sur, F_u_sur, ADRF to different parameters (AOD, SSA, ASY, albedo, columnar water vapor and ozone) in the sensitivity test. The X-axis value shows the ratio of the input to the actual value to dimensionalize the parameters for comparison.**