# Peer review of "Retrieval of Gridded Aerosol Direct Radiative Forcing Based on Multiplatform Datasets"

_Atmospheric Measurement Techniques, 2019_

## Referee Comment (RC1) · Anonymous Referee #1 · 5 Sep 2019

Atmospheric aerosols play a crucial role in regional radiative budgets. The changes in surface radiation induced by aerosols significantly impacts the surface temperature, crop growth and solar energy availability. However, the current assessment of aerosol direct radiative forcing (ADRF) remains highly uncertain. This paper proposes the method for ADRF simulation in East China based on a combination of MODIS and MERRA model reanalysis data. The comparison with surface measurements ensures the accuracy of this retrieval, and it is the highlight of this research along with detailed discussion about error analysis. The analysis of ADRF distribution provides valuable information about the aerosol radiative effect in the heavily polluted region, East China, and in the period when it experienced an unprecedented economic boom. This method is helpful to study aerosol-induced changes in the surface radiation under climate change and agricultural economic impact over China. The research makes an interesting and potentially useful contribution. However, there are several points need to be addressed and revise carefully. Overall, I recommend this manuscript for publication in AMT with Minor Revision.

General comments:

1. The most important factor affecting ADRF simulation is the accuracy of MERRA -2 SSA in East China. I notice that the chosen sunphotometers all locate in the northern China, these sites may not represent the whole East China, what comparison results in other sunphotometers of East China?

2. It seems chaotic in the error analysis of ADRF. In section 4.2, the author analyses the error sources between simulated F_d_sur and the observations (cloud contamination, different spatial and temporal representativeness. . ..). Is this associated with ADRF errors? In section 4.4, the author also conducts the uncertainty analysis of different aerosol optical parameters. The difference between this should be clarified, more explanations and structure adjustment are also needed.

3. In Line 313, "the magnitude of ADRF is higher than the most cities in the world, such as Spain (Esteve et al., 2014), Gasan (Kim et al., 2006) and Karachi (Alam et al., 2011)". It is suggested to add some more discussions about the comparison between East China and other countries. For example, why is ADRF in East China higher than other countries?

Specific comments:

1. Line 68: "Furthermore, aerosol microphysical parameters are crucial in ADRF simulation, including single scattering albedo and the asymmetry parameter (ASY), but their retrieval remains challenging." It is suggested to adjust word order for readability, for example, it is better to be written: "Furthermore, the retrieval of aerosol microphysical parameters remains challenging, including single scattering albedo and the asymmetry

parameter (ASY)."

2. Line 161: "Before ADRF simulation, one of the inputs, SSA from MERRA-2, was evaluated firstly". Why does the author evaluate SSA first in the Results? Please make some more explanation about the goal of the SSA evaluation at the beginning of the Results.

3. Line 276: "MERRA-2 SSA is always underestimated in these conditions, potentially leading to the negative errors in the simulated F_d_sur". Why?

4. There are some mistakes in grammar in this paper, please check them carefully. For example, Line 19: "has" should be "have"; Line 57 "restricted" should be "are restricted"; Line 92: "including" should be "includes".

---

## Referee Comment (RC2) · Anonymous Referee #2 · 1 Oct 2019

**General comment**

In this study, the SBDART radiative transfer model was implemented over East China, using as input satellite data from MODIS and reanalysis data from MERRA-2, NCEP and ECMWF. Measurements from ground stations were used to validate some of the input data and the output. In terms of methodological approach, the study lacks in innovation, since both the radiative transfer model and the input data have been widely used in the past. However, the authors claim that this is the first time that this apporach is implemented over a large area in East China, for the 16-year period 2000-2016. In such a case, the study would benefit if the analysis and discussion regarding the model output, and possible relevant explanations, were expanded. This would lead to a better contribution of this study to the existing literature regarding aerosol loads over China, their effects and changes during the previous years, and I strongly encourage the authors to expand the study in this direction. Overall, I recommend reconsideration of this study after major revisions.

**Specific comments**

Abstract

The authors use three stations to validate their results. How much representative are these stations regarding the entire East China region studied?

1 Introduction

Lines 53-55: The statement regarding the different levels of aerosol cooling effects over different areas of China renders the question of representativeness of the three stations used here for validation purposes very important: do the three sites capture the variability in aerosol types and sources (and consequently optical properties) over East China well enough?

Lines 62-63: I understand that the authors want to highlight the advantages of satellite-based aerosol retrievals. The result, however, is misleading and should be complemented with some of the disadvantages. For example, "continuous temporal coverage" is hardly achieved from satellite observations, since it depends e.g. on satellite orbits and the presence of clouds.

Lines 1-2: "… have rarely been addressed…": Please mention these few studies.

Line 3: This disadvantage of satellite measurements is true globally, not only over China.

Line 90: As with MERRA-2 and MODIS before, please mention here also the data set used for the gridded aerosol vertical profiles.

2 Data

Lines 100-104: Please mention that these results regard previous MODIS AOD collections and update with relevant studies using collection 6.

Line 105: "… at a wavelength of 0.55 $\mu$m". How is SSA treated spectrally?

Lines 120-122: Please be more specific: was the daily MCD43C3 albedo product used? (this is mentioned in Table 2, but it should also be mentioned here). Which band(s)? Which measure is the "confidence index" and which values were selected to ensure accuracy?

Lines 128-144: The aerosol vertical profile plays indeed an important role in the corresponding forcing calculations, but the way that it was estimated and incorporated in the radiative transfer calculations is not clear: what was the default of the radiative transfer model and what changes were implemented? Were the calculations described here performed in this study or in the references provided? Please provide references for the WRF Model and NCEP-FNL algorithm. Please also give more details on the output of these calculations and how it was used in the radiative transfer model.

Lines 141-144: Please mention what kind of interpolation was used for the spatial resolution homogenization. The authors should also provide relevant information on the temporal resolution. As mentioned in Table 2, the AOD and TOA fluxes are instantaneous (although it should also be mentioned that they are available once per day), and other data sets are hourly and daily. What was the temporal resolution of the radiative transfer calculations?

Table 2: To my knowledge, the spatial resolution of the daily surface albedo product MCD43C3 is 0.05°×0.05°, not 0.2°×0.2°.

3 Methodology

Please provide more details on the radiative transfer calculations: were they spectral or broadband? Which solar spectrum was used as input? How was the spectral variation of aerosol properties and surface albedo treated?

4 Results and discussion

4.1 Retrieval of aerosol properties

Lines 163-164: What do the authors mean by "other sites in East China did not have enough data for analysis"? SSA is a crucial and highly uncertain parameter in the calculation of aerosol radiative effects, and in my opinion, every quality-screened sunphotometer data, even of short ranges or intermittent, would add to the credibility of the SSA reanalysis data used here.

Line 179: Do the authors claim that SSA values are similar throughout the study region? This would be intriguing considering the size of the study region (10°×14°) and the high variability of aerosol sources within it. Perhaps an analysis of SSA spatial variability based on MERRA-2 data would clarify this issue.

Last paragraph of Sect. 4.1: The approach used to restrict ASY values described here is interesting and promising. However, it implies that all other parameter values (except ASY) are correct and do not affect the difference between estimated and measured F_u_toa: the authors practically assume that varying ASY only is enough to match F_u_toa values, and the ensuing ASY value can then be trusted. This assumption can deviate from reality if differences

between real and retrieved values of other parameters (e.g. SSA, AOD) occur. The authors should include a discussion on this issue and its possible consequences. Additionally, a description of the statistics of ASY values retrieved here would also be helpful and informative.

4.2. Validation of the method

Line 216: "… in the single grid…" Do the authors mean the three grids of corresponding stations? Please rephrase.

Line 221: Please be more specific and give details regarding the performed quality control.

Line 231: I don't understand how the authors reach to this conclusion based on Fig. 5. The fitting lines suggest that the simulated $F\_d\_sur$ is overestimated in low values and underestimated in high values. The range of values could easily be explained by e.g. the seasonal variation in solar zenith angle, rather than different pollution levels. Even if pollution levels were the only explanation for this range, low $F\_d\_sur$ values should be related to polluted conditions, since more aerosols would block larger parts of the radiation reaching the surface.

Line 232: What do the authors mean with the term "smooth"? Please explain.

Line 235: "… especially in clear conditions". Again, low values of $F\_d\_sur$ are somehow associated with clear conditions. Please explain.

Line 236: "… southern and northern sites of East China…". Based on Fig. 1, Fuzhou and Yong'an are in the southern sites of the study region, however Baoshan is more central than northern.

Lines 244-246: How is the presence of clouds inferred from the MODIS true color map?

Lines 270-283: It is not clear what the authors claim here regarding the effect of aerosol origin on ADRF. What is the difference between the northward and southward directions and how does this difference explain the different error sign? If I understand correctly, the authors claim that aerosols from northward directions are mainly anthropogenic and strongly scattering. What about the southward directions? If aerosols originate at sea, aren't they also strongly scattering? Please discuss more and clarify.

4.3 Long-term ADRF retrieval in East China

The authors mention in this subsection many names of places. It would be helpful for the reader to have these places shown on a map.

Lines 297-298: This explanation is interesting. Do the authors mean that AOD values are similar between northern and southern areas, and the large differences in forcing should be attributed to the aerosols in the North being more scattering? Comparing the maps shown in Fig. 6a with corresponding spatial distributions of AOD and SSA could clarify this point.

Line 300: "locates" should read "located", and "it" before "blocks" should be omitted. Please also rephrase the end of this sentence: are aerosol accumulations higher or lower?

Line 305: "dominated natural aerosols": please rephrase. "Weaker cooling effect": is this due to lower concentrations or different optical properties? Please clarify.

Lines 308-311: "In addition… measurements". There are many grammatical errors in this sentence that need correction. Furthermore, past tense should be used here. More important, however, is the fact that this is a very significant finding of the study, and it should be further investigated here. What kind of changes did the authors find? What where the differences between North and South? The 16-year long data sets used as input are adequate enough to investigate possible reasons for the changes found in the model output, and could provide useful insights. Hence, I do not agree with the statement that this result "needs to be further identified and explored with additional measurements". This is an important part of the analysis that should be included here.

Lines 312-313. Please provide possible explanations for these patterns. Again, comparisons with input data and relevant studies could give useful insights.

4.4 Sensitivity test and uncertainty analysis

Lines 349-340: I do not understand how the sensitivity test presented here can lead to this conclusion regarding the aerosol profiles. Please clarify.

5 Conclusion

Lines 383-389: Some of the findings presented in previous sections are repeated here. They sould rather be summarized.

**Technical corrections**

Line 20: please replace "Terra and Aqua" with "Terra and Aqua MODIS".

Line 32: please omit "with" and "the" in "climate change".

Line 38: Liao et al. should read "2015".

Line 43: Is this a global average value?

Line 52: Nyeki et al. should read "2015".

Line 56: please add "the" before "wider knowledge".

Line 57: please add "are" after "measurements".

Line 60: Qiu et al should read "2017".

Line 65: "Graaf" should read "de Graaf".

Lines 77-78: Please replace "Levet" with "Levelt" and "Tilstra et al." with "Tilstra and Stammes".

Line 78: Please consider replacing "undesirable" with a more appropriate term.

Line 84: Please replace "After SSA determined, ASY, the only unknow inputs" with "After SSA is determined, ASY, the only unknown input".

Line 87: Please replace "propose" with "provide" and "in the clear sky" with "under clear skies".

Lines 88-89: Please consider rephrasing. Furthermore, East China is the study area, rather than the "validation area".

Line 92: Please replace "including" with "includes".

Line 93: Please replace "was" with "is".

Line 94: Please add "is" after "method".

Lines 150-151: Please correct the ECMWF acronym (also in Fig. 2).

Lines 179-180: There is no "Che et al., 2017" study in the references.

Line 182: Buchard et al. should read "2017".

Line 192: "Chang, 2013" is not included in the references.

Line 212: Please add "be" before "input".

Line 215: Please omit "to" before "applied".

Line 217: "was" should be replaced by "were".

Line 287: Please add "the" before "past".

Line 307: Please omit "of".

Line 308: "the positive value of ADRF can occur especially in the bright surface" should be replaced by "positive values of ADRF can occur especially over bright surfaces".

Line 312: "It reflects ADRF shows…". Please rephrase.

Line 313: Please omit "the" before "most".

Line 314: The Alam et al., 2011 citation is not included in the references.

Line 317: Please replace "with combining of" with "combined with".

Line 318: Do you mean "Wu et al., 2016"?

Line 341: Guan et al. should read "2010".

Line 370: Please correct the ECMWF acronym.

Line 380: Please replace "additionally" with "additional".

Line 382: Please include "of" after "validation".

---

## Author Comment (AC1) · 5 Nov 2019

**Reply to Anonymous Referee #1 (amt-2019-311-AC1):**

Atmospheric aerosols play a crucial role in regional radiative budgets. The changes in surface radiation induced by aerosols significantly impacts the surface temperature, crop growth and solar energy availability. However, the current assessment of aerosol direct radiative forcing (ADRF) remains highly uncertain. This paper proposes the method for ADRF simulation in East China based on a combination of MODIS and MERRA model reanalysis data. The comparison with surface measurements ensures the accuracy of this retrieval, and it is the highlight of this research along with detailed discussion about error analysis. The analysis of ADRF distribution provides valuable information about the aerosol radiative effect in the heavily polluted region, East China, and in the period when it experienced an unprecedented economic boom. This method is helpful to study aerosol-induced changes in the surface radiation un-der climate change and agricultural economic impact over China. The research makes an interesting and potentially useful contribution. However, there are several points need to be addressed and revise carefully. Overall, I recommend this manuscript for publication in AMT with Minor Revision.

**Response:**

We appreciate the positive comments of the Referee. It is your valuable comments that make this manuscript more scientific and rational. We have studied the comments carefully and tried our best to revise our manuscript accordingly. Some errors and deficiencies were also revised through our self-check process. We would like to express our thanks for the constructive comments again, and we look forward to hearing your feedback. The specific corrections and comments that are addressed below.

**General comments:**

1. The most important factor affecting ADRF simulation is the accuracy of MERRA -2 SSA in East China. I notice that the chosen sunphotometers all locate in the northern China, these sites may not represent the whole East China, what comparison results in other sunphotometers of East China?

**R:** Thanks for the insight comments and I totally agree with the reviewer's opinion. Here, we found all

the SSA data from the open source (AERONET, http://aeronet.gsfc.nasa.gov/). All sunphotometer sites over East China were collected to validate with MERRA-2 SSA. Six sites of East China were chosen, that is, Xuzhou, Shouxian, Hefei, Taihu, Pudong and Hangzhou. The detail information of site locations and the comparisons between MERRA-2 and sunphotometer SSA are shown in Table 3 and Figure 3. The validation results also have been analyzed:

*"The location of the sunphotometers was shown in Figure 3(a), and their geographical characteristics, observation periods, sample numbers as well as the fitted regression equation between MERRA-2 and sunphotometer SSA were presented in Table 3. In East China, six sunphotometer sites, Xuzhou (117.14ºE, 34.22ºN), Shouxian (116.78ºE, 32.56ºN), Hefei (117.16ºE, 31.91ºN), Taihu (120.22ºE, 31.42ºN), Pudong (121.79ºE, 31.05ºN) and Hangzhou (120.16ºE, 30.29ºN) (Figure 3a), were chosen for comparison with MERRA-2 SSA data. Table 3 shows the locations of these sunphotometers, and their geographical characteristics, observing periods as well as fitted regression equation between MERRA-2 and sunphotometer SSA. The detailed comparisons at Xuzhou, Shouxian and Hefei were shown in the Figure 3b. Orange dots represent Xuzhou samples and orange line is the according fitting curve, while green represents Shouxian, and black is Hefei. Figure 3c displays the comparison results at Taihu, Pudong and Hangzhou. Red denotes Taihu, purple is Pudong and yellow is Hangzhou. As shown in Figure 3, dashed lines are the range of ±10% relative error, all samples in Taihu, Pudong and Hefei, 94% of samples in Xuzhou, 93% in Shouxian and 98% in Hangzhou fall within the ±10% error. This finding suggests that MERRA-2 SSA agrees well with the sunphotometer data, even though few SSA samples are beyond the error range. Furthermore, the slopes of linear fitting curve are less than 1 at all sites except Shouxian (Table 3), and it reveals that MERRA-2 SSA has systematic biases at most area of East China."*

*Table 3: The geographical characteristics, observing period, sample number of sunphotometer sites. The fitted regression equations between MERRA-2 and sunphotometer SSA are also shown here. In the equation, x represents SSA sample, y represents fitted value of SSA.*

| Location | Lon/Lat | Observing period | Sample number | Fitted regression equation between MERRA-2 and sunphotometer SSA |
|---|---|---|---|---|
| Xuzhou (Urban) | 117.14°E/34.22°N | 2013.8-2016.12 | 514 | y=0.02+0.94x |
| Shouxian (Rural) | 116.78°E/32.56°N | 2008.5-2008.12 | 26 | y=-0.45+1.46x |
| Hefei (Urban) | 117.16°E/31.91°N | 2005.11-2005.12 2008.1-2008.11 | 19 | y=0.09+0.85x |
| Taihu (Rural) | 120.22°E/31.42°N | 2005.1-2012.12 2015.1-2016.12 | 230 | y=0.2+0.75x |
| Pudong (Urban) | 121.79°E/31.05°N | 2010.12-2012.10 2014.1-2015.11 | 84 | y=0.49+0.46x |
| Hangzhou (Urban) | 120.16°E/30.29°N | 2008.4-2009.2 | 45 | y=0.38+0.57x |

*Figure 3: (a) The location of six sunphotometer sites over East China. (b) The scatter plots of SSA between MERRA-2 and sunphotometer in Xuzhou, Shouxian and Hefei. Orange dots represent Xuzhou samples and orange line is the fitting curve of Xuzhou samples while green represents Shouxian and black represents Hefei. Dashed lines are the range of ±10% relative error. (c) The scatter plots of SSA between MERRA-2 and sunphotometer in Taihu, Pudong and Hangzhou. Red dots represent Taihu samples and red line is the fitting curve of Taihu samples while purple denotes Pudong and yellow is Hangzhou. Dashed lines are the range of ±10% relative error.*

2. **It seems chaotic in the error analysis of ADRF. In section 4.2, the author analyses the error sources between simulated F_d_sur and the observations (cloud contamination, different spatial and temporal representativeness. Is this associated with ADRF errors? In section 4.4, the author also conducts the uncertainty analysis of different aerosol optical parameters. The difference between this should be clarified, more explanations and structure adjustment are also needed.**

**R:** We thanks for your insight comments. The object of Section 4.2 is F_d_sur and Section 4.3 is ADRF. Section 4.2 discusses the specific validation cases, and the error sources have indeed impact on ADRF. However, the goal of uncertainty analysis in Section 4.4 is to quantify how much uncertainty of ADRF induced by the inputs (AOD, SSA, albedo, etc.). In general, the object of two sections are different, and for clarification, the uncertainty analysis in Section 4.4 has been moved to Section 4.3 in the revised manuscript.

3. **In Line 313, "the magnitude of ADRF is higher than most cities in the world, such as Spain (Esteve et al., 2014), Gasan (Kim et al., 2006) and Karachi (Alam et al., 12011)". It is suggested to add some more discussions about the comparison between East China and other countries. For example, why is ADRF in East China higher than other countries?**

**R:** Thanks for your suggestion. The more discussions about the comparison between East China and other countries have been added in the revised manuscript:

*"The main reason is that AOD in East China is much larger than these cities, since East China has experienced rapid urbanization and economic development in the past 17 years and AOD is much larger than these regions. For example, mean AOD in East China is 0.62 in this study during 2003-2011 while AOD is 0.19 in Spain during 2003-2011 (Esteve et al., 2014)."*

**R:** Thanks for your careful suggestions. The according mistakes have been corrected, and some other grammar errors were also revised in the manuscript through our self-check process.

[revised manuscript text omitted]

H̶The hourly SSA product was provided by MERRA-2. w̶a̶s̶ ̶e̶s̶t̶i̶m̶a̶t̶e̶d̶ ̶b̶y̶ ̶t̶h̶e̶ ̶r̶a̶t̶i̶o̶ ̶o̶f̶ ̶t̶o̶t̶a̶l̶ ̶a̶e̶r̶o̶s̶o̶l̶ ̶s̶c̶a̶t̶t̶e̶r̶i̶n̶g̶ ̶a̶e̶r̶o̶s̶o̶l̶ ̶o̶p̶t̶i̶c̶a̶l̶ ̶t̶h̶i̶c̶k̶n̶e̶s̶s̶ ̶(̶A̶O̶T̶)̶ ̶t̶o̶ ̶t̶o̶t̶a̶l̶ ̶a̶e̶r̶o̶s̶o̶l̶ ̶e̶x̶t̶i̶n̶c̶t̶i̶o̶n̶ ̶A̶O̶T̶ ̶a̶t̶ ̶a̶ ̶w̶a̶v̶e̶l̶e̶n̶g̶t̶h̶ ̶o̶f̶ ̶0̶.̶5̶5̶ ̶μ̶m̶.̶ MERRA-2 combines GEOS-5 and the three-dimensional variational data assimilation (3DVar) Gridpoint Statistical Interpolation analysis system (GSI). GEOS-5 is coupled to the Goddard Chemistry, Aerosol, Radiation and Transport (GOCART) aerosol module, which includes five particulate species (sulfate, dust, sea salt, organic and black carbon) (Colarco et al., 2010). The optical properties of these aerosols are primarily from the Optical Properties of Aerosols and Clouds (OPAC) dataset (̶H̶e̶s̶s̶ ̶e̶t̶ ̶a̶l̶.̶,̶ ̶1̶9̶9̶8̶)̶, in which aerosol optical parameters are calculated based on the microphysical data (size distribution and spectral refractive index) under the assumption of spherical particles and they are given for up to 61 wavelengths between 0.25 and 40 μm (Hess et al., 1998).t̶h̶e̶ S̶S̶A̶ ̶v̶a̶l̶u̶e̶ ̶a̶t̶ ̶0̶.̶5̶5̶ ̶μ̶m̶ ̶c̶a̶n̶ ̶b̶e̶ ̶i̶n̶t̶e̶r̶p̶o̶l̶a̶t̶e̶d̶ ̶a̶t̶ ̶t̶h̶e̶ ̶o̶t̶h̶e̶r̶ ̶w̶a̶v̶e̶l̶e̶n̶g̶t̶h̶s̶.̶ 
[revised manuscript text omitted]

**Figure S2.**  MODIS Terra true color map composed by 1, 4, and 3 channels on October 18, 2014 (https://worldview.earthdata.nasa.gov/).

**Figure S3.**  The response of downward radiative fluxes at the surface (F_d_sur), upward radiative fluxes at the surface (F_u_sur), aerosol direct radiative forcing (ADRF) to different parameters (AOD, SSA, ASY, albedo, columnar water vapor and ozone) in the sensitivity test.

**Figure S4.** The occurrence frequency of annual ADRF for each grid cell in the North and South of East China during 2000-2016.

[Figure]

990

**Figure S1. The boxplot of MERRA-2 SSA and sunphotometer in Pudong, Taihu, and Xuzhou. The central marks in each box are the median value while the lower and upper edges of the boxes indicate 25th and 75th percentiles. The whiskers show extreme values and the outliers are maked with "+".**

995

[Figure]

**Figure S1. Sketch map of aerosol vertical profile (He et al., 2008). Two-layer aerosol model is characterized by aerosol well-mixed in the PBL and exponential decay of the aerosol extinction coefficient with altitude above the top of PBL.**

[Figure]

**Figure S2. 48 h backward trajectories of air mass by HYSPLIT 4, which are terminating at Fuzhou at 500m altitude level. Blue lines are the trajectories with negative relative error and the red lines are the tragectories with positive relative error.**

[Figure]

**Figure S2. MODIS Terra true color map composed by 1, 4, and 3 channels on October 18, 2014 (https://worldview.earthdata.nasa.gov/). The red rectangle box (40*40km) is the MODIS AOD average window in Baoshan pyranometers site.**

1005

[Figure]

**Figure S3. 48 h backward trajectories of air mass arriving at Yong'an at 500m altitude level and calculated every 24 h from October 22 to October 24, 2015. The start time is 2:00 (UTC) during satellite passing by.**

[Figure]

**Figure S3. The response of downward radiative fluxes at the surface (F_d_sur), upward radiative fluxes at the surface (F_u_sur), aerosol direct radiative forcing (ADRF) to different parameters (AOD, SSA, ASY, albedo, columnar water vapor and ozone) in the sensitivity test. The X-axis value shows the ratio of the input to the actual value to dimensionalize the parameters for comparison.**

[Figure]

**Figure S4. The occurrence frequency of annual ADRF for each grid cell in the North and South of East China during 2000-2016.**

---

## Author Comment (AC2) · 5 Nov 2019

**Reply to Anonymous Referee #2 (amt-2019-311-AC2):**

**General comment**

**In this study, the SBDART radiative transfer model was implemented over East China, using as input satellite data from MODIS and reanalysis data from MERRA-2, NCEP and ECMWF. Measurements from ground stations were used to validate some of the input data and the output. In terms of methodological approach, the study lacks in innovation, since both the radiative transfer model and the input data have been widely used in the past. However, the authors claim that this is the first time that this approach is implemented over a large area in East China, for the 16-year period 2000-2016. In such a case, the study would benefit if the analysis and discussion regarding the model output, and possible relevant explanations, were expanded. This would lead to a better contribution of this study to the existing literature regarding aerosol loads over China, their effects and changes during the previous years, and I strongly encourage the authors to expand the study in this direction. Overall, I recommend reconsideration of this study after major revisions.**

**Response:**

Great thanks for the valuable comments and it's our honor to get these constructive suggestions. New relevant analysis has been incorporated in the revised paper, including:

1. The detailed analysis about the spatiotemporal changes of ADRF in East China based on past 16 years ADRF data.
2. The relationship between ADRF (output) and aerosol optical parameters (input) was discussed, and possible reasons for the changes of ADRF also been represented.
3. The data information and methodology also have been displayed more detailly.
4. For readability and clarification, we adjusted the structure of the manuscript, revised and edited some literally-uninterpretable sentences.

In addition, some errors and deficiencies were also revised through our self-check process. We would like to express our thanks for the constructive comments again, and we look forward to hearing your feedback. The detailed modifications have been included in a new supplement material document.

**Specific comments**
**Abstract**

**The authors use three stations to validate their results. How much representative are these stations regarding the entire East China region studied?**

**R:** Thanks for your insightful comment, and this is a good question. We attempt to have much complete validation but limited observation is a great challenge. With accordance to close relationship between ADRF and AOD&SSA, the validation part should include representative sites with different aerosol properties as far as possible. In our study, three sites (Baoshan, Fuzhou, and Yong'an) were used to validate the ADRF simulation. The aerosol concentrations in the urban and suburb sites are obvious. Baoshan and Fuzhou are typic urban sites, which can represent the situation with relative high concentrations of aerosols. Yong'an is suburb with low concentrations of aerosols. Meanwhile, the location of sites determines the aerosol sources. Baoshan and Fuzhou are near the sea, and aerosol types are similar with the aerosols in other coastal areas of East China. Yong'an is inland, and it can represent the inland situation of East China. With proper consideration of aerosol properties, we think the validation is representative in our study.

The according discussion has been addressed in Section 4.2 in the revised manuscript:

*"Baoshan and Fuzhou are urban and coastal sites while Yong'an represents suburb and inland sites. The different aerosol concentration levels and abundant aerosol types in these sites can represent the most of aerosol properties in East China."*

**1 Introduction**

**Lines 53-55: The statement regarding the different levels of aerosol cooling effects over different areas of China renders the question of representativeness of the three stations used here for validation purposes very important: do the three sites capture the variability in aerosol types and sources (and consequently optical properties) over East China well enough?**

**R:** Many thanks for your question and perhaps I didn't write it clearly in Lines 53-55. My meaning is that the different levels of aerosol cooling effects over different areas of "China", not "East China". We all know that China is such a vast area and aerosol properties have huge difference between west and east

China. However, in East China, aerosol optical properties have much smaller difference compared with those in China. Baoshan and Fuzhou are urban and coastal stations while Yong'an represents suburb and inland station. The different aerosol concentration levels and abundant aerosol types in these stations can represent the most of aerosol properties in East China. For clarification and simple, we have deleted the misleading statement *"and these measurements imply that aerosols exert different levels of cooling effect near the surface in different regions"* in the Introduction.

**Lines 62-63: I understand that the authors want to highlight the advantages of satellite-based aerosol retrievals. The result, however, is misleading and should be complemented with some of the disadvantages. For example, "continuous temporal coverage" is hardly achieved from satellite observations, since it depends e.g. on satellite orbits and the presence of clouds.**

**R:** We thank you for the careful review and the sentence (Line 62-63) has been rephrased:

*"Compared to the above methods, satellite remote sensing has an outstanding advantage of delivering aerosol information with higher spatial resolution and larger spatial coverage."*

**Lines 1-2: "... have rarely been addressed...": Please mention these few studies.**

**R:** Thanks for the specific comments and we have added the according information about few studies about surface ADRF distribution in the revised manuscript:

*"Thus far, long-term estimates of the surface ADRF distribution have rarely been addressed, and few studies gave a full picture of surface ADRF over land (e.g.: Thomas et al., 2013; Chung et al., 2016)."*

*Reference:*

*Chung, C. E., Chu, J. E., Lee, Y., Van Noije, T., Jeoung, H., Ha, K. J., and Marks, M.: Global fine-mode aerosol radiative effect as constrained by comprehensive observations, Atmos. Chem. Phys.,16(13), 8071-8080, https://doi.org/10.5194/acp-16-8071-2016,2016.*
*Thomas, G. E., Chalmers, N., Harris, B., Grainger, R. G., and Highwood, E. J.: Regional and monthly and clear-sky aerosol direct radiative effect (and forcing) derived from the GlobAEROSOL-AATSR satellite aerosol product, Atmos. Chem. Phys., 13(1), 393-410, https://doi.org/10.5194/acp-13-393-2013, 2013.*

**Line 3: This disadvantage of satellite measurements is true globally, not only over China.**

**R:** We agree with your opinion, and the according sentence *"especially in China, one of the most populated and polluted regions globally"* has been deleted in the revised manuscript.

**Line 90: As with MERRA-2 and MODIS before, please mention here also the data set used for the gridded aerosol vertical profiles.**

**R:** We appreciate you for the careful review and have already added the dataset information used for the aerosol profiles in the revised manuscript:

*"In our study, aerosol vertical profiles are determined by the Weather Research and Forecasting Model (WRF, version 3.2.1) and the National Centers for Environmental Prediction-Final Operational Global Analysis (NCEP-FNL). The detailed algorithm of aerosol profiles can be found in Section 2".*

**2 Data**

**Lines 100-104: Please mention that these results regard previous MODIS AOD collections and update with relevant studies using collection 6.**

**R:** Thanks for your careful suggestion. The previous MODIS AOD collection information and the update of C6 have been added in the revised manuscript:

*"Compared with C5, MODIS C6 mainly updated the cloud mask to allow heavy smoke retrievals and fine-tuned the assignments for aerosol types as function of season and location over the land. Levy et al. (2013) made a comparison between MODIS C5, C6 and AERONET, and found that the correlation coefficient of C6/AERONET increases slightly, and the slope and offset of the regression curve only changed slightly compared with C5/AERONET."*

R: Thanks for your careful comments. Daily MCD43C3 albedo product was used. The band is shortwave (0.3-5µm). The "confidence index" includes the data quality information and it quantifies the proportion

of the data inversion retrieval in each pixel. For example, confidence index 0 denotes the best quality (100% with full inversion and no fill values), this index increases with the decrease of the proportion, and 4 denotes 50% or less fill values. Here, the albedo values with high quality bit index (0-4) were used. In the revised manuscript, more information about albedo product has been added:

*"Another important parameter for ADRF simulations is the surface albedo, and it was derived from the daily MODIS MCD43C3 black-sky albedo product (C6). Surface albedo product includes seven narrow bands and three broadbands (visible (0.3-0.7μm), near-infrared (0.7-5.0μm), and SW (0.3-5μm)). Here, albedo product in SW band was used in our study. Each file contains 16 days of combined Level 3 data from the satellites Aqua and Terra, with a spatial resolution of 0.05°. It also contains the data quality information, that is, the proportion of inversion retrieval information in each pixel. For example, data quality index 0 represents the best quality (100% with full inversion and no fill values), this index increases with the decrease of the proportion of inversion retrieval pixel, and 4 represents 50% or less fill values. Notably, to ensure the accuracy, only the albedo values with high quality index (0-4) were used."*

**Lines 128-144: The aerosol vertical profile plays indeed an important role in the corresponding forcing calculations, but the way that it was estimated and incorporated in the radiative transfer calculations is not clear: what was the default of the radiative transfer model and what changes were implemented? Were the calculations described here performed in this study or in the references provided? Please provide references for the WRF Model and NCEP-FNL algorithm. Please also give more details on the output of these calculations and how it was used in the radiative transfer model.**

R: Thank for your advice and we are sorry for the unclear introduction about aerosol vertical profile. More details about aerosol vertical profile have been added in the Data:

*"In SBDART, aerosol vertical profile is shaped by aerosol density and the according altitude. The aerosol density is a proportion of AOD in different altitude, and the overall profile is scaled by AOD. The aerosol density is set to fall exponentially between two altitudes by default. In our study, aerosol vertical profile in SBDART was derived from two-layer aerosol vertical distribution model, which is proposed by He et al. (2008). In this two-layer aerosol model (Figure S1), aerosol extinction coefficient is assumed to*

*decrease exponentially with altitude above the top of the planet boundary layer (PBL) and the extinction coefficient keeps uniform below the PBL. Based on this aerosol model, two inputs of aerosol vertical profile need to be determined, PBL and aerosol layer height (ALH). ALH is defined as the level where the aerosol extinction coefficient decreases to 1/e (scale height) of that at the top of PBL. PBL and ALH input to SBDART along with the according aerosol density. In this study, PBL was simulated by a three-domain, two-way nested simulating of the Weather Research and Forecasting Model (WRF, version 3.2.1). ALH can be influenced by the transport of air mass and the convective dispersion of aerosols, both of which are usually associated with large-scale weather systems. Based on the different meteorological conditions, an automated workflow algorithm of ALH was constructed, and ALH can be estimated by the meteorological parameters (relative humidity, temperature, wind speed and wind direction) from the National Centers for Environmental Prediction-Final Operational Global Analysis (NCEP-FNL). The detailed algorithm and the according calculations of PBL and ALH retrieval can be found in the He et al. (2016). The aerosol profiles were utilized to calculate the surface-level visibility from MODIS/ AOD, the long-term spatial comparison with surface measurement displays that correlation coefficients of 90% samples are greater than 0.6, and 68% of the samples have coefficients higher than 0.7 (He et al., 2016)."*

**R:** Thanks for your careful reviews. The information regarding spatial interpolation and spatial resolution have been added in the revised manuscript:

*"In this study, bilinear interpolation was used in these datasets, and these datasets were interpolated to a spatial resolution of 0.1°×0.1° to collocate with MODIS/AOD data. For temporal resolution, AOD and TOA radiation fluxes were from the MODIS and CERES sensor aboard the Terra satellite respectively, and they are available once per day. Both SSA and ERA-Interim are hourly means, surface albedo product in daily means. The ADRF simulations were only performed at the passing over of the Terra satellite under clear skies."*

**Table 2: To my knowledge, the spatial resolution of the daily surface albedo product MCD43C3 is 0.05″0.05°, not 0.2″0.2°.**

R: Sorry to make a mistakes about the spatial resolution of MCD43C3, MCD43C3 resolution has been corrected as 0.05° ×0.05° in the revised manuscript.

**3 Methodology**

**Please provide more details on the radiative transfer calculations: were they spectral or broadband? Which solar spectrum was used as input? How was the spectral variation of aerosol properties and surface albedo treated?**

R: Thanks for your useful comments. The broadband surface irradiance was simulated by radiative transfer model. Here, LOWTRAN 7 solar spectrum was adopted in SBDART. SBDART also includes the standard aerosol models derived from Shettle and Fenn (1975), in which aerosol optical parameters are wavelength dependence and the scattering parameters depend on the surface relative humidity. Users can also define different aerosol parameters in different wavelength. The default of the according spectral information is interpolated/extrapolated to all wavelengths using linear fitting on SSA/ASY, and using Ångstrom coefficients on AOD. According to Wang et al. (2009), it has very minor effect on the accuracy of irradiance simulation using spectrally averaged values of aerosol parameters compared with detail spectral information. Therefore, aerosol parameters at 0.55 μm were used in the radiative transfer model. As for surface albedo, it is simply assumed that angular distribution of surface-reflected radiation is completely isotropic in the model. Five basic surface types (ocean water, lake water, vegetation and snow) can be used to parameterize the spectral surface albedo, and users can also specify the mixture ratio of

these types and spectral (or uniform spectral) albedo. Here, MODIS SW MCD43C3 (0.3-5 μm) product is used as albedo input, and it is nearly consistent with wavelength coverage (0.25-4 μm) of the output surface irradiances in the radiative transfer model. The according modification has been added in the revised manuscript:

*"In this study, SBDART model was used to estimate broadband SW (0.25-4 μm) surface irradiances and ADRF over East China. It is on the basis of the DISORT radiative transfer model, the low-resolution band models developed for LOWTRAN 7 atmospheric transmission, and the Mie scattering results for light scattering by water droplets and ice crystals (Ricchiazzi et al., 1998). Here, LOWTRAN 7 solar spectrum was adopted in SBDART. This radiative transfer model also includes the standard aerosol models derived from Shettle and Fenn (1975), in which aerosol optical parameters are wavelength dependence and the scattering parameters depend on the surface relative humidity. Users can also define different aerosol parameters in different wavelength. The default of the according spectral information is interpolated/extrapolated to all wavelengths using linear fitting on SSA/ASY, and using Ångstrom coefficients on AOD. According to Wang, P. et al. (2009), the input of aerosol parameters has very minor effect on the accuracy of irradiance simulation when using spectrally averaged values compared with detail spectral information. Therefore, aerosol parameters (AOD, SSA, ASY) at 0.55 μm were used in the radiative transfer model. As for surface albedo, it is simply assumed that angular distribution of surface-reflected radiation is completely isotropic in the model. In our study, MODIS SW MCD43C3 (0.3-5 μm) product is used as albedo input, and it is nearly consistent with wavelength coverage (0.25-4 μm) of the output surface irradiances in SBDART."*

**R:** Thanks for your constructive review and we totally agree that all sunphotometers over East China can be used to validate with MERRA-2 SSA. In the revised manuscript, six sites of East China were chosen, that is, Xuzhou, Shouxian, Hefei, Taihu, Pudong and Hangzhou. The detail information of site locations is shown in Table 3, and the comparisons between MERRA-2 and sunphotometer SSA are displayed in Figure 3. The validation results also have been analyzed:

*"In East China, six sunphotometer sites, Xuzhou (117.14ºE, 34.22ºN), Shouxian (116.78ºE, 32.56ºN), Hefei (117.16ºE, 31.91ºN), Taihu (120.22ºE, 31.42ºN), Pudong (121.79ºE, 31.05ºN) and Hangzhou (120.16ºE, 30.29ºN) (Figure 3a), were chosen for comparison with MERRA-2 SSA data. The location of the sunphotometers was shown in Figure 3(a), and their geographical characteristics, observation periods, sample numbers as well as the fitted regression equation between MERRA-2 and sunphotometer SSA were presented in Table 3. The detailed comparisons at Xuzhou, Shouxian and Hefei were shown in Figure 3b. Orange dots represent Xuzhou samples and orange line is the according fitting curve, while the green represents Shouxian, and the black is Hefei. Figure 3c displays the comparison results at Taihu, Pudong and Hangzhou. Red denotes Taihu, the purple is Pudong and the yellow is Hangzhou. As shown in Figure 3, dashed lines are the range of ±10% relative error, all samples in Taihu, Pudong and Hefei, 94% of samples in Xuzhou, 93% in Shouxian and 98% in Hangzhou fall within the ±10% error. This finding suggests that MERRA-2 SSA agrees well with the sunphotometer data, even though few SSA samples are beyond the error range. Furthermore, the slopes of linear fitting curve are less than 1 at all sites except Shouxian (Table 3), and it reveals that MERRA-2 SSA has systematic biases at most area of East China."*

*Table 3: The geographical characteristics, observing period, sample number of sunphotometer sites. The*

*fitted regression equations between MERRA-2 and sunphotometer SSA are also shown here. In the equation, x represents SSA sample, y represents fitted value of SSA.*

| Location | Lon/Lat | Observing period | Sample number | Fitted regression equation between MERRA-2 and sunphotometer SSA |
|---|---|---|---|---|
| Xuzhou (Urban) | 117.14°E/34.22°N | 2013.8-2016.12 | 514 | y=0.02+0.94x |
| Shouxian (Rural) | 116.78°E/32.56°N | 2008.5-2008.12 | 26 | y=-0.45+1.46x |
| Hefei (Urban) | 117.16°E/31.91°N | 2005.11-2005.12 2008.1-2008.11 | 19 | y=0.09+0.85x |
| Taihu (Rural) | 120.22°E/31.42°N | 2005.1-2012.12 2015.1-2016.12 | 230 | y=0.2+0.75x |
| Pudong (Urban) | 121.79°E/31.05°N | 2010.12-2012.10 2014.1-2015.11 | 84 | y=0.49+0.46x |
| Hangzhou (Urban) | 120.16°E/30.29°N | 2008.4-2009.2 | 45 | y=0.38+0.57x |

*Figure 3: (a) The location of six sunphotometer sites over East China. (b) The scatter plots of SSA between MERRA-2 and sunphotometer in Xuzhou, Shouxian and Hefei. Orange dots represent Xuzhou samples and orange line is the fitting curve of Xuzhou samples while green represents Shouxian and black represents Hefei. Dashed lines are the range of ±10% relative error. (c) The scatter plots of SSA between MERRA-2 and sunphotometer in Taihu, Pudong and Hangzhou. Red dots represent Taihu samples and*

*red line is the fitting curve of Taihu samples while purple denotes Pudong and yellow is Hangzhou. Dashed lines are the range of ±10% relative error.*

**Line 179: Do the authors claim that SSA values are similar throughout the study region? This would be intriguing considering the size of the study region (10°´14°) and the high variability of aerosol sources within it. Perhaps an analysis of SSA spatial variability based on MERRA-2 data would clarify this issue.**

**R:** We guess the reviewer may misunderstand that "SSA values are similar throughout the study region", my original meaning is that mean value of MERRA-2 SSA in three sites (Pudong, Taihu and Xuzhou) is consistent with the surface measurements in these areas (Line 178-179). For clarify and simple, we have deleted them in the revised manuscript.

**Last paragraph of Sect. 4.1: The approach used to restrict ASY values described here is interesting and promising. However, it implies that all other parameter values (except ASY) are correct and do not affect the difference between estimated and measured F_u_toa: the authors practically assume that varying ASY only is enough to match F_u_toa values, and the ensuing ASY value can then be trusted. This assumption can deviate from reality if differences between real and retrieved values of other parameters (e.g. SSA, AOD) occur. The authors should include a discussion on this issue and its possible consequences. Additionally, a description of the statistics of ASY values retrieved here would also be helpful and informative.**

**R:** We are thankful for the insightful comment. Varying ASY only is enough to match F_u_toa when the other parameter values (e.g. AOD, SSA) is accurate. As the reviewer's suggested, we have added a detailed discussion about this assumption, and the statistics of retrieved ASY have also been included in the revised manuscript:

*"Following this method, ASY was retrieved in each grid cell over East China. The range of retrieved ASY is 0.50-0.80, and the mean ASY is 0.63, which is consistent with the observation site (Taihu) in East China (Xia et al., 2007). According to Mie theory, ASY is determined by the size distribution and the complex*

*refractive index of aerosols. Therefore, the difference of ASY in East China can be partly related with the difference of fine mode radius. Xia et al. (2007) has reported that the fine mode volume median radius at Taihu site averages 0.181 μm over a range of AOD from 0.6-1.0, while it is 0.168 μm in northern China. In ASY retrieval, ASY is assumed to vary enough to match F_u_toa with ensuring the accuracy of all other inputs (e.g. AOD, SSA). This assumption can deviate from the reality if there are obvious differences between real and retrieval values of other inputs. This above condition can easily occur in the process of ASY retrieval, when ASY cannot be retrieved (ASY=NaN). Even if ASY can be obtained, ASY can be inaccurate when other inputs have large biases. The uncertainty of ASY caused by the other inputs (AOD, SSA, albedo, CERES F_u_toa) will be quantified in the following uncertainty analysis (Section 4.3)."*

**Line 231: I don't understand how the authors reach to this conclusion based on Fig. 5. The fitting lines suggest that the simulated F_d_sur is overestimated in low values and underestimated in high values. The range of values could easily be explained by e.g. the seasonal variation in solar zenith angle, rather than different pollution levels. Even if pollution levels were the only explanation for this range, low F_d_sur values should be related to polluted conditions, since more aerosols would block larger parts of the radiation reaching the surface.**

**R:** We totally agree with your opinion and there are some mistakes about the interpretation of low/high simulated F_d_sur in Line 231. The range of F_d_sur can be mainly due to the seasonal variation in solar zenith angle. The according discussions in Line 230-Line 235 has been deleted and revised in the manuscript.

**Line 232: What do the authors mean with the term "smooth"? Please explain.**

**R:** My original meaning "this method can smooth F_d_sur variations" is that this method makes the range of simulated F_d_sur smaller than the observed F_d_sur. But it seems like this word "smooth" is inappropriate to be used here, so this sentence has been deleted in the revised manuscript.

**Line 235: "... especially in clear conditions". Again, low values of F_d_sur are somehow associated with clear conditions. Please explain.**

**R:** We totally agree with your suggestions and I am apologized for this wrong discussion about the simulated F_d_sur. These sentences have been deleted in the revised manuscript.

**Line 236: "... southern and northern sites of East China...". Based on Fig. 1, Fuzhou and Yong'an are in the southern sites of the study region, however Baoshan is more central than northern.**

**R:** We agree that and the statement is not accurate. The according sentence has been modified according to the characteristics of these sites:

*"Nevertheless, satisfactory comparison results indicate the suitability and feasibility of ADRF retrieval over East China in the off/near the sea and urban/suburb sites of East China, although the types of underlying surface and aerosol properties are evidently different in these areas."*

**Lines 244-246: How is the presence of clouds inferred from the MODIS true color map?**

**R:** Thanks for your comment. Clouds are easily identified, because the cloud is white in the MODIS true color map. Compared with sunphotometers, MODIS AOD can be overestimated easily when the cloud exists. Here, to make this better readable, an example of MODIS true color map was shown in the revised manuscript and the according description has also added in the revised manuscript:

*"Meanwhile, the olive green dots denote the specific case in which the site is completely covered by clouds inferred from the MODIS true color map composed by channels 1, 4 and 3. Taking one olive green cases (Baoshan, October 18, 2014) for an example. As shown in the Figure S2, it is obvious that a large amount of cloud exists in the area of 29°N-31°N and 120°E-122°E, and Baoshan site is at the edge of the cloud. In this case, MODIS AOD was overestimated compared with sunphotometer AOD, this because some cloud effects were not completely removed from the MODIS/AOD calculation. Therefore, a large discrepancy can occur in these cases between simulated F_d_sur and observation."*

[Figure]

*Figure S2. MODIS Terra true color map composed by 1, 4, and 3 channels on October 18, 2014 ([https://worldview.earthdata.nasa.gov/](https://worldview.earthdata.nasa.gov/)). The red rectangle box (40\*40km) is the MODIS AOD average window in Baoshan pyranometers sites.*

**Lines 270-283: It is not clear what the authors claim here regarding the effect of aerosol origin on ADRF. What is the difference between the northward and southward directions and how does this difference explain the different error sign? If I understand correctly, the authors claim that aerosols from northward directions are mainly anthropogenic and strongly scattering. What about the southward directions? If aerosols originate at sea, aren't they also strongly scattering? Please discuss more and clarify.**

**R:** Thanks for your valuable suggestions. We admit that, in our last manuscript, aerosols from north and south are different degree of scattering, but it cannot explain the different error sign. As reviewer says, both anthropogenic aerosols from northward direction and sea salt aerosols from southward direction are strongly scattering. Therefore, the discussion regarding the relationship between aerosol origin and error sign did not make sense. Therefore, this according sentences (Line 270-283) have been deleted in the revised manuscript.

**4.3 Long-term ADRF retrieval in East China**

**The authors mention in this subsection many names of places. It would be helpful for the reader to have these places shown on a map.**

**R:** Thanks for your careful comment. All the mentioned places, including lakes and mountains, have been added in the map (Figure 1).

[Figure]

*Figure 1: The map of research area, topography, major lakes and mountains in East China. The red circles denote the locations of three pyranometers (Baoshan, Fuzhou and Yong'an). This figure was generated by ArcGIS, version 10.2. Map source: Map World (National Platform for Common Geospatial Information Services, [www.tianditu.gov.cn](www.tianditu.gov.cn) ).*

**Lines 297-298: This explanation is interesting. Do the authors mean that AOD values are similar between northern and southern areas, and the large differences in forcing should be attributed to the aerosols in the North being more scattering? Comparing the maps shown in Fig. 6a with corresponding spatial distributions of AOD and SSA could clarify this point.**

**R:** Thanks for your very insightful suggestion. The comparison results between the mean spatial distributions of ADRF, AOD and SSA are shown in the Figure 7. The according explanation about the spatial pattern of ADRF also has been added in the revised manuscript:

*"According to the uncertainty analysis, the spatial pattern of ADRF is closely associated with the inputs (SSA and AOD). Based on this, comparison was conducted among the mean spatial distribution of ADRF, AOD and SSA during 2000-2016 (Figure 7). It is clear to see that ADRF pattern is very similar to the negative phase of AOD pattern, that is, the areas of high AOD have low ADRF. As for SSA, the higher value can be found in the South than the North, which indicating the aerosols in the South are generally*

*more scattering than the North. Therefore, the large difference between North and South can be mainly attributed to the difference in AOD. The industry locations and topography between the North and South are obviously different. With the development of economy and urbanization, large amounts of anthropogenic aerosols in the North can impose strong cooling radiative effect in the past two decades. It is worth noting that, although western Shandong has lower urbanization compared with YRD, aerosol cooling effect in western Shandong is even larger than in YRD. This is because Yimeng mountain (these mentioned places are all shown in Figure 1) located in the middle of Shandong, blocks the west flow, leading to the enhancement of the aerosol accumulations and high AOD near its western border (He et al., 2012b). Meanwhile, Shandong is also easily impacted by air pollution transported from North China. In addition, high absolute value of ADRF is also found in Poyang Lake in Jiangxi with abundance of anthropogenic aerosols, and these areas are surrounded by the mountains, the poor ventilation condition makes aerosols enhanced. Compared with the North, the South is characterized by more extensive vegetation coverage and less human activities, and AOD is relatively lower in the South (Figure 7b) and aerosols have weaker cooling effect.''*

[Figure]

*Figure 7: Averaged spatial distribution of (a)ADRF (unit: W m⁻²), (b)AOD and (c)SSA during 2000-2016 in the East China.*

*Reference:*

*He, Q., Li, C., Geng, F., Lei, Y., and Li, Y.: Study on long-term aerosol distribution over the land of East China using MODIS data, Aerosol Air Qual. Res.12,300-315, https://doi.org/10.4209/aaqr.2011.11.0200, 2012b.*

**Line 300: "locates" should read "located", and "it" before "blocks" should be omitted. Please also rephrase the end of this sentence: are aerosol accumulations higher or lower?**

**R:** Thanks for your careful suggestion, and the sentences in Line 300 has been rephrased in the revised manuscript:

*"This is because Yimeng mountain (these mentioned places are all shown in Figure 1) located in the middle of Shandong, blocks the west flow, leading to the enhancement of the aerosol accumulations and high AOD near its western border."*

**Line 305: "dominated natural aerosols": please rephrase. "Weaker cooling effect": is this due to lower concentrations or different optical properties? Please clarify.**

**R:** We thank for your suggestion. The sentence (Line 305) has been revised. The weaker cooling effect is due to lower concentrations and lower AOD. The modification has been added in the revised manuscript:

*"Compared with the North, the South is characterized by more extensive vegetation coverage and less human activities, and AOD is relatively lower in the South (Figure 7b) and aerosols have weaker cooling effect."*

**Lines 308-311: "In addition… measurements". There are many grammatical errors in this sentence that need correction. Furthermore, past tense should be used here. More important, however, is the fact that this is a very significant finding of the study, and it should be further investigated here. What kind of changes did the authors find? What where the differences between North and South? The 16-year long data sets used as input are adequate enough to investigate possible reasons for the changes found in the model output, and could provide useful insights. Hence, I do not agree with the statement that this result "needs to be further identified and explored with additional measurements". This is an important part of the analysis that should be included here.**

**R:** Great thank for your suggestion. The grammatical errors have been corrected. In addition, more analysis has been added in the revised manuscript, including the spatiotemporal changes of ADRF, the difference between North and South, and the reasons for the changes. This important part of analysis goes

as follows:

[revised manuscript text omitted]
 weaker of aerosol cooling effect in this region. However, a few regions experience the decrease of ADRF especially in the northeast and south area of Yimeng mountain in Shandong. In general, the changes of ADRF in the past 17 years are mainly due to the anthropogenic emissionsin East China. In addition, Paulot et al. (2018) further pointed that there is a nonlinear relationship between anthropogenic emissions and AOD/ADRF when considering the mix and oxidation of different emissions.''*

[Figure]

*Figure 6: Yearly mean ADRF distributions during 2000-2016 over East China (unit: W m$^{-2}$).*

[Figure]

*Figure 7: Averaged spatial distribution of (a)ADRF (unit: W m-2), (b)AOD and (c)SSA during 2000-2016 in the East China.*

[Figure]

*Figure 8: The time series of monthly mean ADRF (blue) and AOD (red) in East China from 2000 to 2016. Dashed lines represent the Mann-Kendell (MK) fitting trend of ADRF and AOD.*

[Figure]

*Figure 9: The spatial distribution of ADRF trend in East China during 2000-2016 (unit: W m-2 month-1). Hatched regions represent those exceeding the 90% significance level.*

[Figure]

*Figure 8: The time series of monthly mean ADRF (blue) and AOD (red) in East China from 2000 to 2016. Dashed lines represent the Mann-Kendell (MK) fitting trend of ADRF and AOD.*

[Figure]

*Figure 9: The spatial distribution of ADRF trend in East China during 2000-2016 (unit: W m-2 month-1). Hatched regions represent those exceeding the 90% significance level.*

Line 192: "Chang, 2013" is not included in the references.

Line 212: Please add "be" before "input".

Line 215: Please omit "to" before "applied".

Line 217: "was" should be replaced by "were".

Line 287: Please add "the" before "past".

**Line 307: Please omit "of".**

**Line 308: "the positive value of ADRF can occur especially in the bright surface" should be replaced by "positive values of ADRF can occur especially over bright surfaces".**

**Line 312: "It reflects ADRF shows…". Please rephrase.**

**Line 313: Please omit "the" before "most".**

**Line 314: The Alam et al., 2011 citation is not included in the references.**

**Line 317: Please replace "with combining of" with "combined with".**

**Line 318: Do you mean "Wu et al., 2016"?**

**Line 341: Guan et al. should read "2010".**

**Line 370: Please correct the ECMWF acronym.**

**Line 380: Please replace "additionally" with "additional".**

**Line 382: Please include "of" after "validation".**

**R:** Great thanks for the careful and useful suggestions. These above Technical corrections have been modified in the revised manuscript one by one.

[revised manuscript text omitted]

995    **Figure S1. Sketch map of aerosol vertical profile (He et al., 2008). Two-layer aerosol model is characterized by aerosol well-mixed in the PBL and exponential decay of the aerosol extinction coefficient with altitude above the top of PBL.**

[Figure]

**Figure S2. 48 h backward trajectories of air mass by HYSPLIT 4, which are terminating at Fuzhou at 500m altitude level. Blue lines are the trajectories with negative relative error and the red lines are the tragectories with positive relative error.**

[Figure]

**Fig. S2. MODIS Terra true color map composed by 1, 4, and 3 channels on October 18, 2014 (https://worldview.earthdata.nasa.gov/). The red rectangle box (40*40km) is the MODIS AOD average window in Baoshan pyranometers site.**

[Figure]

1010 Figure S3. 48 h backward trajectories of air mass arriving at Yong'an at 500m altitude level and calculated every 24 h from October 22 to October 24, 2015. The start time is 2:00 (UTC) during satellite passing by.

[Figure]

1015     **Figure S3. The response of downward radiative fluxes at the surface (F_d_sur), upward radiative fluxes at the surface (F_u_sur), aerosol direct radiative forcing (ADRF) to different parameters (AOD, SSA, ASY, albedo, columnar water vapor and ozone) in the sensitivity test. The X-axis value shows the ratio of the input to the actual value to dimensionalize the parameters for comparison.**

[Figure]

1020

**Figure S4. The occurrence frequency of annual ADRF for each grid cell in the North and South of East China during 2000-2016.**

---

## Referee Report (RR1)

**General comment**

The authors have replied to all comments and taken into account all suggestions for additional analyses. The paper is improved to a large extent, due both to the added discussions, as well as the re-structured content. I recommend acceptance of the manuscript after the following minor revisions:

**Specific comments**

(line numbers refer to the marked-up version of the revised manuscript)

Lines 107-111: the added discussion on MODIS C6 should be moved to the end of this paragraph, i.e. after the discussion on validation studies concerning C5.

Lines 235-237: these two sentences belong to the figure caption and should be removed from the main text.

Lines 420-421: "AOD is much larger than these cities/regions" is repeated in the same sentence. Please rephrase.

Line 471: Figure 8 shows that AOD and ADRF are anti-correlated. Shouldn't a negative value of the correlation coefficient be expected?

Figure 8: Please mention in the figure caption that these data are deseasonalized. The legend is also probably wrong (continuous red line should be AOD?), and the red dashed line (AOD fitting trend) seems to match exactly the zero line. Please check.

---

## Author Response (AR2)

**General comment**

**The authors have replied to all comments and taken into account all suggestions for additional analyses. The paper is improved to a large extent, due both to the added discussions, as well as the re-structured content. I recommend acceptance of the manuscript after the following minor revisions:**

**Response:**

We appreciate these positive comments, it is your valuable comments that make this manuscript improved. The specific corrections that are addressed below.

**Specific comments**

**(line numbers refer to the marked-up version of the revised manuscript)**

**Lines 107-111: the added discussion on MODIS C6 should be moved to the end of this paragraph, i.e. after the discussion on validation studies concerning C5.**

**R:** We thanks for your insight comments. The modification has been made in Lines 107-111:

*"To acquire ADRF, the inputs (aerosol optical depth (AOD), SSA, ASY, albedo, etc.) to the radiative transfer model were determined from a combination of satellite and reanalysis datasets. AOD was derived from Collection 6 (C6) of MODIS Level 2 products over land (10-km resolution at the nadir) from the Terra satellite (Levy et al., 2013). MODIS AOD retrieval primarily employs three spectral channels, centered at 0.47, 0.66, and 2.1 μm and is interpolated at 0.55 μm (Kaufman et al., 1997). Li et al. (2003) demonstrated that the MODIS AOD Level 2 product is appropriate in eastern China and exhibits high precision. Compared with C5, MODIS C6 mainly updated the cloud mask to allow heavy smoke retrievals and fine-tuned the assignments for aerosol types as function of season and location over the land. Levy et al. (2013) made a comparison between MODIS C5, C6 and AERONET, and found that the correlation coefficient of C6/AERONET increases slightly, and the slope and offset of the regression curve only changed slightly compared with C5/AERONET. In addition, He et al. (2010) found that MODIS AOD was highly correlated with sunphotometer (CE318) measurements at 7 sites in the Yangtze River Delta (YRD) region (118°-123°E, 29°-*

*33°N), with a correlation coefficient of 0.85 and with 90% of cases falling in the range of ΔAOD = ± 0.05 ± 0.20 AOD (Chu et al., 2002). Thus, the uncertainty in the AOD is regarded as 20% in this study."*

*Reference:*

*Chu, D. A., Kaufman, Y. J., Ichoku, C., Remer, L. A., Tanré, D., and Holben, B. N.: Validation of MODIS aerosol optical depth retrieval over land, Geophys. Res. Lett., 29(12), 1617−1621, https://doi.org/10.1029/2001gl013205, 2002.*

*He, Q., Li, C., Tang, X., Li, H., Geng, F., and Wu, Y.: Validation of MODIS derived aerosol optical depth over the Yangtze River Delta in China, Remote Sens. Environ. 114(8), 1649-1661, https://doi.org/10.1016/j.rse.2010.02.015, 2010.*

*Kaufman, Y. J., Tanre, D. L., Remer, A., Vermote, E. F., Chu, A., and Holben, B. N.: Operational remote sensing of tropospheric aerosol over land from EOS moderate resolution imaging spectroradiometer, J. Geophys. Res., 102,17051–17067, https://doi.org/10.1029/96jd03988, 1997.*

*Levy, R. C., Mattoo, S., Munchak, L. A., Remer, L. A., Sayer, A. M., and Hsu, N.C.: The Collection 6 MODIS aerosol products over land and ocean, Atmos. Meas. Tech. 6, 2989-3034, https://doi.org/10.5194/amtd-6-159-2013, 2013.*

*Li, C., Mao, J., Lau, A., Yuan, Z., Wang. M., and Liu, X.: Characteristics of distribution and seasonal variation of aerosol optical depth in Eastern China with MODIS products (in Chinese). Chin. Sci. Bull. 48(22),2488-2495, https://doi.org/10.1360/03wd0224, 2003.*

**Lines 235-237: these two sentences belong to the figure caption and should be removed from the main text.**

**R:** Thanks for your suggestion. The two sentences about the descriptions of Figure 3 (Lines 235-237) has been removed from the revised manuscript.

**Lines 420-421: "AOD is much larger than these cities/regions" is repeated in the same sentence. Please rephrase.**

**R:** Thanks for your careful advice. The sentence has been rephrased:

*"The main reason is that AOD in East China is much larger than these cities, since East China has experienced rapid urbanization and economic development in the past 17 years and a robust increase can be found in anthropogenic emissions."*

**Line 471: Figure 8 shows that AOD and ADRF are anti-correlated. Shouldn't a negative value of the correlation coefficient be expected?**

**R:** We are sorry to make the mistakes about that. The correlation coefficient of AOD and ADRF is negative, with the value of -0.72. The according modification has been corrected in the revised manuscript.

**Figure 8: Please mention in the figure caption that these data are deseasonalized. The legend is also probably wrong (continuous red line should be AOD?), and the red dashed line (AOD fitting trend) seems to match exactly the zero line. Please check.**

**R:** Thanks for your careful suggestion. The caption in Figure 8 has added that "these data are deseasonalized". The legends have been corrected, that is, continuous red line is AOD, dashed line represent the AOD fitting trend. The red dashed line seems to match the zero line, this is because the AOD trend is about $0.3068 \times 10^{-4}$ month$^{-1}$, this value is nearly close to zero. This modification has been done in the revised manuscript.

[revised manuscript text omitted]